# 50 years of balloon-borne ozone profile measurements at Uccle, Belgium: short history, scientific relevance and achievements in understanding the vertical ozone distribution

Roeland Van Malderen[1], Dirk De Muer[1], Hugo De Backer[1], Deniz Poyraz[1], Willem W. Verstraeten[1], Veerle De Bock[1], Andy W. Delcloo[1], Alexander Mangold[1], Quentin Laffineur[1], Marc Allaart[2], Frans Fierens[3], Valérie Thouret[4]

[1]Royal Meteorological Institute of Belgium, Uccle (Brussels), 1180, Belgium
[2]KNMI, P.O. Box 201, 3730 AE De Bilt, the Netherlands
[3]Belgian Interregional Environment Agency (IRCEL - CELINE), Brussels, 1030, Belgium
[4]Laboratoire d'Aérologie, Université de Toulouse, CNRS, UPS, Toulouse, France

*Correspondence to*: R. Van Malderen (roeland.vanmalderen@meteo.be)

**Abstract.** Starting in 1969, and with three launches a week, the Uccle (Brussels, Belgium) ozonesonde dataset is one of longest and densest of the world. Moreover, as the only major change was the switch from Brewer-Mast (BM) to Electrochemical Concentration Cell (ECC) ozonesonde types in 1997 (when the emissions of ozone depleting substances peaked), the Uccle time series is very homogenous. In this paper, we briefly describe which efforts have been taken during the first three decades of the 50 years of ozonesonde observations to guarantee the homogeneity between ascent and descent profiles, under changing environmental conditions (e.g. $SO_2$), and between the different ozonesonde types. This paper focusses on the 50 years long Uccle ozonesonde dataset and aims to demonstrate its past, present and future relevance to ozone research in two application areas: (i) the assessment of the temporal evolution of ozone from the surface to the (middle) stratosphere, and (ii) as backbone for validation and stability analysis of both stratospheric as well as tropospheric satellite ozone retrievals. Using the Long-term Ozone Trends and Uncertainties in the Stratosphere (LOTUS) multiple linear regression model (SPARC/IO3C/GAW, 2019), we found that the stratospheric ozone concentrations at Uccle declined at a significant rate of around 2% $dec^{-1}$ since 1969, rather consistently over the different stratospheric levels. This overall decrease can mainly be assigned to the 1969-1996 period with a rather consistent decline rate around -4% $dec^{-1}$. Since 2000, a recovery between +1 to +3% $dec^{-1}$ of the stratospheric ozone levels above Uccle is observed, although not significant and not for the upper stratospheric levels measured by ozonesondes. Throughout the entire free troposphere, a very consistent increase of the ozone concentrations at 2 to 3 % $dec^{-1}$ is measured since both 1969 and 1995, the latter trend being in almost perfect agreement with the trends derived from the In-service Aircraft for a Global Observing System (IAGOS) ascent/descent profiles at Frankfurt. As the amount of tropopause folding events in the Uccle time series increased significantly over time, increased stratosphere-to-troposphere transport of recovering stratospheric ozone might partly explain these increasing tropospheric ozone concentrations, despite the levelling off in (tropospheric) ozone precursor emissions and notwithstanding the continued increase of mean surface ozone concentrations. Furthermore, we illustrate the

crucial role of ozonesonde measurements for validation of satellite ozone profile retrievals. With the operational validation of Global Ozone Monitoring Experiment GOME-2, we show how the Uccle dataset can be used to evaluate the performance of a degradation correction for the MetOp-A/GOME-2 UV sensors. In another example, we illustrate that the Microwave Limb Sounder (MLS) overpass ozone profiles in the stratosphere agree within ±5% with the Uccle ozone profiles between 10 and 70 hPa. Another instrument on the same AURA satellite platform, Tropospheric Emission Spectrometer (TES), is generally positively biased with respect to the Uccle ozonesondes in the troposphere by up to ~10 ppbv, corresponding to relative differences up to ~15 %. Using the Uccle ozonesonde time series as reference, we also demonstrate that the temporal stability of those last two satellite retrievals is excellent.

## 1    Introduction

Ozone, $O_3$, is a key trace gas in the Earth's atmosphere, where it mainly resides between the surface and the top of the stratosphere (about 50 km), with the highest concentrations in the lower to middle stratosphere (90% of total column ozone amount). Ozone is mainly produced in the tropical stratosphere and transported to the lower stratosphere at high latitudes. Depending on its altitude, ozone is involved in different chemical reactions and therefore has a different impact on life on Earth. Stratospheric ozone absorbs the harmful solar ultraviolet (UV) radiation, hereby protecting life on Earth, and warming the stratosphere. This protective shield has been in danger due to anthropogenic emissions of ozone depleting substances (ODSs, such as chlorofluorocarbons or CFCs) since the 1970s, with the Antarctic springtime ozone hole as the most striking signature. Thanks to the Montreal Protocol (1987, and subsequent amendments and adjustments), positive trends in the ozone concentrations in the upper stratosphere are observed since 2000 (WMO, 2018, Chapters 3 and 4, and SPARC/IO3C/GAW, 2019). Ozone is also an important absorber of infrared (terrestrial) radiation, mainly in the tropopause region, and therefore can act as a greenhouse gas at certain altitudes, estimated to have contributed ~20% as much positive radiative forcing as $CO_2$ since 1750 (IPCC, 2013). Tropospheric ozone is also the main source of the OH free radical, the primary oxidant in the atmosphere, responsible for removing many compounds (including atmospheric pollutants) from tropospheric air. At the surface, ozone is an air pollutant that adversely affects human health, natural vegetation, and crop yield and quality (e.g. Cooper et al., 2014).

Because of the many roles of ozone, the knowledge and measurement of the vertical distribution of the ozone concentration in the atmosphere – and its variability in time – is crucial. Vertical ozone profiles can be obtained from ground-based instruments (Dobson/Brewer Umkehr, lidar, FTIR, and microwave radiometer), balloon-borne techniques (ozonesondes), and satellite-based measurements (using solar/stellar occultation, limb emission/scattering and nadir-viewing techniques), see e.g. Hassler et al. (2014) for details. In this research, we focus on ozonesondes, lightweight and compact balloon-borne instruments measuring the ozone concentration from the surface through the mid-stratosphere (about 10 hPa or 30 km). In electrochemical ozonesondes atmospheric ozone is measured via an electrochemical reaction of ambient air bubbling in a solution of potassium iodide (KI), by means of a stable miniature pump. In a Brewer-Mast sonde two electrodes of different

metal are immersed in a buffered KI solution (Brewer and Milford, 1960), while Electrochemical Concentration Cell (ECC) sondes consists of two half cells with different solutions of KI as electrodes (Komhyr, 1969). The ozonesonde is launched in tandem with a radiosonde that also transmits air pressure, temperature, humidity and wind data to a ground station. With a 20-30s response time of the ozone cells and an ascent rate of about 6 m s$^{-1}$, the effective vertical resolution of the ozone signal lies nowadays around 150 m. Before the digital sounding systems era the vertical resolution was coarser due to the manual sampling technique by the operator, providing only measurements at significant levels.

Regular measurements with ozonesondes started in the second half of the 1960s at a few sites: in 1965 at Aspendale (Australia, but moved to other suburbs of Melbourne thereafter, i.e. Laverton and Broadmeadows), in 1966 at Resolute Bay (Canada), in 1967 at Hohenpeissenberg (Germany), in 1968 at Payerne (Switzerland) and at Tateno (Tsukuba, Japan), in 1969 at Uccle (Belgium) and Sapporo (Japan), and in 1970 at Wallops Island (USA). These ozone sounding stations provide the longest time series of vertical ozone distribution. Up to an altitude of about 30 km, ozonesondes constitute the most important data source with long-term data coverage for the derivation of ozone trends with sufficient vertical resolution, particularly in the climate sensitive altitude region around the tropopause. Furthermore, ozonesondes are widely used to study photochemical and dynamical processes in the atmosphere or to validate and evaluate satellite observations and their long term stability (Smit and ASOPOS panel, 2014, and references therein).

In this paper, we focus on the ozonesonde measurements at Uccle, covering 50 years, demonstrating its scientific relevance and the major achievements. Ozonesondes are still the only technique able to measure the ozone concentrations from the surface all the way up to the middle stratosphere with very high (absolute) accuracy and vertical resolution. Therefore, they have many application areas in which they are crucial: (i) quantifying the long-term variability in stratospheric and tropospheric ozone, (ii) as backbone for satellite validation, with satellites mostly measuring ozone only in stratosphere or upper troposphere, and (iii) for process studies in stratospheric-tropospheric exchange, and chemical production/destruction of ozone. The strength and uniqueness of the ozonesonde measurements, and in particular of the long-term and very dense Uccle dataset, lie in combining all those different aspects of ozone research. In this paper, we will first give a description of the ozonesonde measurements at Uccle from a historical point of view (Sect. 2) and describe briefly which data processing has been applied to the ozonesonde measurements used in this paper (Sect. 3). In Section 4, we assess the time evolution of ozone at Uccle at different vertical layers against the background of recent findings in ozone variability. The fifth section illustrates the important role of the Uccle data for the validation of satellite ozone retrievals. Finally, in Section 6, concluding remarks and perspectives are given.

## 2 The Uccle ozone measurements: a historical overview

In this section, we give a brief overview of the history of the ozone measurements at Uccle (Brussels, Belgium, 50°48'N, 4°21'E, 100 m asl). We explain why the ozone sounding program was initiated more than 50 years ago and discuss the presence of a period of gaps in the time series (Sect 2.1). We also describe which efforts have been taken during this time

period to guarantee the homogeneity of the time series of ozonesondes between ascent and descent profiles (Sect. 2.2.1), with changing environmental conditions (Sect. 2.2.2), and between different ozonesonde types (Sect. 2.2.3). We only give a brief description here, and refer to all the relevant earlier publications for more details.

## 2.1 The start of the ozone observations

The ozone sounding program at the Royal Meteorological Institute of Belgium (RMI) at Uccle was initiated by Prof. Jacques Van Mieghem, director of RMI from 1962 to 1970. Initially the ozone soundings were not performed out of a concern for possible human influence on the ozone layer, but rather to use ozone as a tracer to study the general air circulation in the troposphere and the lower stratosphere. Therefore, from the beginning it was planned to perform regular ozone soundings three times per week (on Monday, Wednesday, and Friday).

In 1965 and 1966 the first few soundings were performed with Regener chemiluminescent ozonesondes, and these data are still available at the World Ozone and Ultraviolet Radiation Data Centre (WOUDC). A well-known effect of this sonde type is that it shows changes in sensitivity during the ascent trajectory (see e.g. Hering and Dütsch, 1965). For that reason it was decided to switch to Brewer-Mast electrochemical ozonesondes (developed by Brewer and Milford (1960) and commercially produced by the Mast Development Company at Iowa, USA) at RMI from November 1966 onwards. Based on a number of criteria such as continuity of the measurements and how well the preparation of the sondes was documented, it was decided to use the ozone soundings for scientific studies only from 1969 onwards, when Dirk De Muer took over the ozone research at RMI (in July 1969).

In the period from February 1983 to January 1985 there were only a few ozone soundings. This gap in our time series was due to funding reductions. Later on, when the Uccle time series of ozone soundings had proved its scientific value and with the growing concern of a human influence on the ozone layer, the continuation of the soundings became less an issue. In the course of time different radio sounding systems have been used. A major change occurred in 1990 when digital data transmission at high sampling rate was introduced, which allowed a higher vertical resolution of the profiles (not only at significant and standard pressure levels).

To normalize the integrated ozone amount of the ozone soundings (essential for BM ozonesondes, see Sect. 2.2.3), the Dobson spectrophotometer (no. 40, D40) at Uccle was used since July 1971; before that date an interpolation of values from other Dobson stations in the European network was employed. In 1984, the Uccle site was equipped with a single Brewer UV spectrophotometer (no. 16), and with a double Brewer instrument (no. 178) in September 2001, to provide total ozone column measurements.

## 2.2 Challenges

### 2.2.1 Frequency response of the electrochemical ozonesonde

In 1970 the ozone sounding program was adapted to gather also the data during the descent of the sonde after balloon burst. De Muer (1981) found that the measured ozone concentrations in the lower stratosphere and the troposphere were systematically higher during the descent than during the ascent (see Fig 1, left panel). Two possible explanations were mentioned: (i) a contamination of the ozonesonde at the ascent (e.g. by reducing constituents in the atmospheric boundary layer, see Sect. 2.2.2) and/or (ii) the response time of the sensor. To investigate the latter, De Muer and Malcorps (1984) analysed the frequency response of the combined ozone sensor and air sampling system of Brewer-Mast ozonesondes by means of a Fourier analysis. They found three different time constants: (i) a first-order process with a time constant of about 17 to 25 s (depending on the solution temperature) caused by the formation of iodine in the solution, (ii) a time constant of 7s, likely to be caused by the diffusion of iodine molecules to the platinum cathode, and (iii) a time constant of about 2.8 min that was explained by another diffusion process, i.e. an adsorption and subsequent desorption process of ozone at the surface of the air-sampling system. The slow first-order process with a time constant of about 20-25 min (found by Salzman and Gilbert, 1959, and taken up by Vömel et al., 2020, and Tarasick et al., 2021) could not be identified, probably because the impact of this process for a 0.1% KI solution would be too small (being 10% of the fast process for a 1% KI solution), as noted in De Muer and Malcorps (1984). With these findings and time constants, a method for deconvolution of the ozone profiles through a process of Fast Fourier transform was developed, and an example of an ozone profile before and after deconvolution is also shown in Figure 1. After deconvolution the observed ozone values during the descent are still larger than the ascent values in the troposphere and the lowest layers in the stratosphere, which was then attributed to the effect of $SO_2$ on the ozonesonde measurements in the boundary layer.

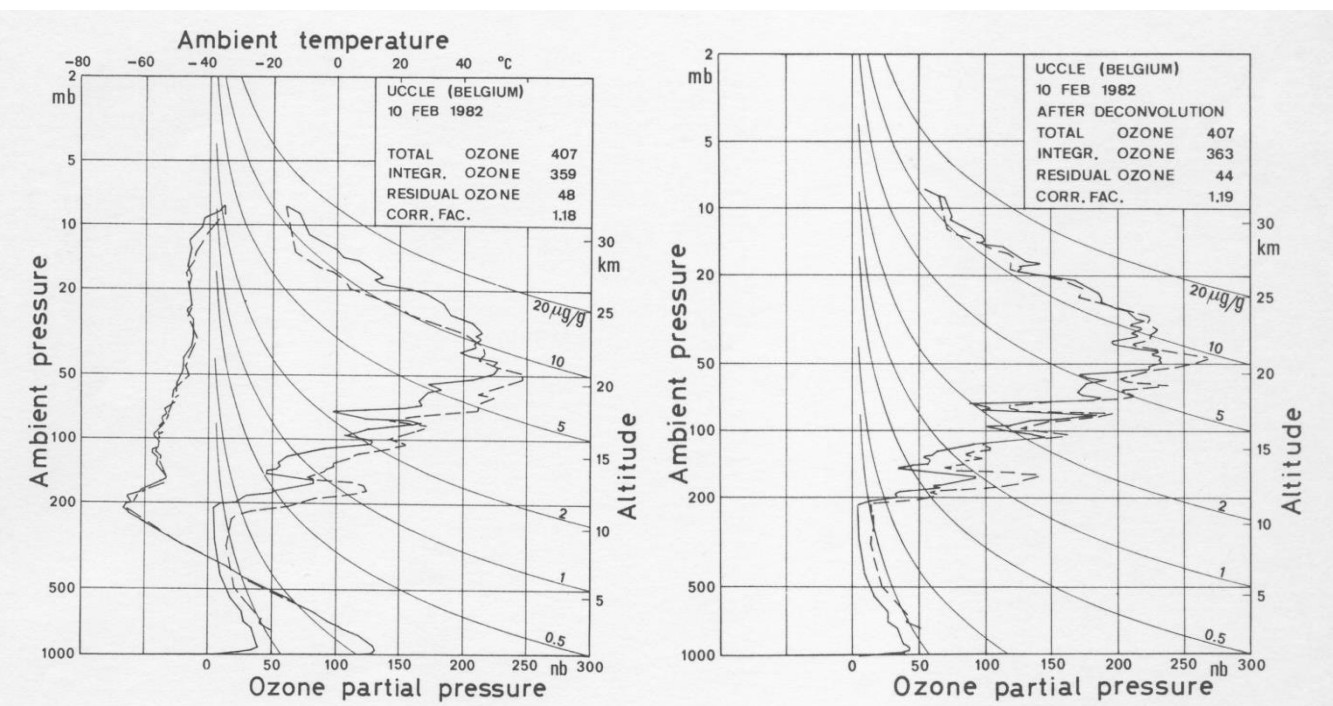

**Figure 1: Ozone sounding at Uccle on 10 February 1982 with a Brewer-Mast ozonesonde before (left) and after (right) deconvolution of the ozone profile for both ascent (solid line) and descent (dashed line) of the sonde. In the left panel, the vertical profile of the air temperature is also shown (figure taken from De Muer & Malcorps, 1984).**

### 2.2.2    The impact of the boundary layer $SO_2$ concentrations on the ozone measurements

Ozonesonde measurements by the KI method are sensitive to interference by oxidizing or reducing agents (e.g. Tarasick et al., 2021, and references therein). In particular, one $SO_2$ molecule cause a reverse current of two electrons, reducing the electrochemical cell response on a 1:1 basis, and excess $SO_2$ can accumulate in the cathode solution, affecting ozonesonde measurements well above the polluted boundary layer (Komhyr, 1969, De Muer and De Backer, 1993, see also Fig.1 and Fig. S1) or near volcanic sites (Morris et al., 2010). Furthermore, in case of a considerable total vertical $SO_2$ column amount, the Dobson total ozone amounts might be overestimated as $SO_2$ has even stronger absorption bands than ozone in the UV 305-340 nm wavelength range used for the total ozone determination (Komhyr and Evans, 1980). As a matter of fact, in the suburban area of Uccle, the $SO_2$ densities near the ground were quite elevated at the start of the ozone measurements, but showed a steep decrease from the late 1960s to the early 1990s (Fig. S2).

As a consequence, the variation of $SO_2$ density near the ground has a twofold effect on ozone soundings with electrochemical sondes: (i) the integrated ozone amount of the (BM) soundings is normalized by means of spectrophotometer data, so that a trend in the latter data will lead to an effect on ozone trends from soundings, and (ii) due to the $SO_2$ interference with the ozonesonde cell reactions, any trend of $SO_2$ causes a distortion of ozone profile trends as a function of altitude.

To minimize this double impact of $SO_2$ on the ozonesonde ozone measurements, two corrections were developed. Based on the comparison between quasi-simultaneous total ozone observations at Uccle with a Dobson and a Brewer spectrophotometer (De Backer and De Muer, 1991), a model connecting $SO_2$ column readings with long-term surface $SO_2$ monitoring measurements was able to subtract a fictitious trend in the Dobson. Applying this correction made the Dobson total ozone trend consistent with both the Brewer trend and the one derived from reprocessed Total Ozone Mapping

Spectrometer (TOMS) satellite data for the sub-periods in which both datasets were available (De Muer and De Backer, 1992). Furthermore, a method to calculate the vertical $SO_2$ distribution associated with each ozone sounding was developed, based on two assumptions: a constant $SO_2$ mixing ratio from the ground to the mixing layer height, and an exponentially decreasing mixing ratio above the mixing layer balancing the integrated $SO_2$ amount to the reduced thickness of the $SO_2$ layer (De Muer and De Backer, 1993). The effect of those two corrections for $SO_2$ interference on the vertical ozone trends

in the 1969-1996 Brewer-Mast period is illustrated in Fig. S3. It shows that those corrections are essential in assessing the trends in tropospheric ozone at Uccle until the mid nineties.

### 2.2.3 The transition from BM to ECC sondes

As mentioned before, at the start of the operational ozone sounding series, the Brewer-Mast sensor type was used. This type of ozonesonde had several issues at that time: (i) a strong reduction of the efficiency of the pump at low pressure (De Backer

et al., 1998a), (ii) the loss of ozone in the sensor itself, causing a relatively high (up to 20%) underestimation of the integrated ozone from a sounding profile with respect to the total ozone measured with a Dobson or a Brewer spectrophotometer, and (iii) a variable response in the troposphere, depending on preparation (Tarasick et al., 2002). Therefore, in the middle of the 1990s, RMI investigated the switch from the BM sondes to the ECC sensors (Komhyr, 1969), which seemed to perform better and were easier to prepare before launch. To document this transition, dual soundings were

launched about twice a month during one year. The comparison between both sensor types on those dual soundings is shown in Fig S4. If standard correction methods for both sensors are used, large statistically significant differences appear: Brewer-Mast sensors overestimate tropospheric ozone and underestimate stratospheric ozone, mainly due to the standard normalisation by linear scaling of the vertical ozone profile for BM sondes. Therefore, De Backer et al. (1998a,b) developed one PRESsure and Temperature dependent Total Ozone normalization (now called PRESTO, see Van Malderen et al., 2016)

correction method for both ozonesonde types, based on (i) measurements of the pump efficiencies of both ozonesonde types in a pressure chamber at Uccle, (ii) a pre-flight comparison of every ozonesonde with a calibrated ozone source in the lab, and (iii) the comparison with the total ozone column measured with the co-located ozone spectrophotometer (full practical details are available in De Backer, 1999). This method is still the operational one at Uccle and has been used to process all the ozonesonde data here (see Section 3). By applying this method, the differences between the dual ozone sounding profiles

are reduced below 3% almost throughout the whole profile and below the statistical significance level (see again Fig. S4). The impact of this new pump correction method on the vertical ozone trends is also significant, especially for the 1969-1996 BM period (see Fig. S3 and also for other periods in Van Malderen et al., 2016).

Further validation of the method was done by comparing the profiles with measurements from the SAGE II satellite instrument (Lemoine and De Backer, 2001). This study showed that the PRESTO correction removed the jump, caused by the BM to ECC transition, in the difference time series with SAGE II at low pressures (compare Fig. 1 and 2 in Lemoine and De Backer, 2001).

## 3    The Uccle ozonesonde dataset

In this paper, the PRESTO correction has been applied to the entire ozonesonde dataset, i.e. to both the BM and ECC ozonesonde types, but with the appropriate different measured pump efficiency coefficients at Uccle for both types, to ensure consistency over the entire data record of 50 years. Although a total ozone normalization is not required for the ECC sonde measurements (Smit and ASOPOS panel, 2014), it is applied for the entire Uccle time series within the PRESTO correction. To calculate the residual ozone above the balloon burst level, we use a combination of the constant mixing ratio approach and the climatological mean obtained from satellite ozone retrievals (McPeters and Labow, 2012). An alternative, homogenized, Uccle ozonesonde corrected dataset is available by request from the authors for the ECC time series since 1997 (Van Malderen et al., 2016), following the principles of the Ozonesonde Data Quality Assessment (O3S-DQA) activity (Smit et al., 2012), but is not used here to maintain the consistency over the entire time series. Differences between both versions of corrected Uccle ECC ozonesonde data, in comparison with the nearby ozonesonde site De Bilt (the Netherlands, 175 km north of Uccle), are highlighted in Van Malderen et al. (2016).

For the BM ozonesondes, the applied PRESTO corrections include (i) a correction for $SO_2$ interference on the ozone soundings (imperative to have reliable lower-tropospheric ozone trend estimates for the period 1969-1996, see Fig. S3), (ii) a correction for a negative background current caused by impurities in the sensor before October 1981, (iii) a correction for box temperatures depending on the insulating capacity of the Styrofoam boxes (a short discussion of those additional corrections and the proper references are given in the appendix A of Van Malderen et al., 2016), and (iv) an altitude correction for VIZ/Sippican radiosonde pressure measurements based on comparisons with wind-finding radar. Without this altitude correction, sonde altitudes were too low up to 1000m at an altitude of 30km, so that the calculated ozone concentrations with VIZ radiosondes were too low by 7.5 to 14%, depending on the manufacturing series of radiosondes (De Muer and De Backer, 1994). Since 1990, the ozonesondes were combined at Uccle with Vaisala RS80 radiosondes, which showed a much smaller difference of the calculated altitude with respect to wind-finding radar data. Therefore, for the digital era period since 1990, no radiosonde pressure sensor bias corrections have been applied, although they have been identified in different studies (e.g. De Backer, 1999; Steinbrecht et al., 2008; Stauffer et al., 2014; Inai et al., 2015).

## 4 Temporal evolution of the vertical ozone concentrations at Uccle

As ozonesondes are the only devices that are able to measure ozone concentrations from the surface up to the middle stratosphere with high vertical resolution, they are very suitable to study and relate the temporal variability of ozone in different atmospheric layers. The evaluation of the temporal variability of the ozone measurements at Uccle is therefore organized in different sections. We first describe the stratospheric (Sect. 4.1) and tropospheric (Sect. 4.2) ozone trends. The relation to other co-located ozone measurements is described in the appendices. Total ozone trends are treated in Annex A and the temporal behaviour of surface ozone and several ozone depleting substances is discussed in Annex B.

### 4.1 Stratospheric ozone trends

For calculating the vertical distribution of trends in the stratospheric ozone concentrations from the Uccle ozonesonde data, we use the altitude relative to the tropopause height as the vertical coordinate. The tropopause applied here is the standard (first) thermal tropopause (WMO, 1957), and is derived from the vertical temperature profiles measured by the Uccle radiosondes, as described in Van Malderen and De Backer (2010). The implemented statistical model to calculate trends is the Long-term Ozone Trends and Uncertainties in the Stratosphere (LOTUS) multiple linear regression (MLR) model (SPARC/IO3C/GAW, 2019). This model uses an independent linear trend (ILT) method as trend term, which is based on two different, independent, trends to describe the ozone decrease until 1997 (ODS increase) and the slow ozone increase since the early 2000s (after the turnaround in ODS concentrations). These two periods have been used since WMO (2014) and their use avoids endpoint anomalies near the turnaround in 1997 for the two independent linear trend terms in the ILT method. Additionally, the LOTUS regression includes two orthogonal components of the quasi biennial oscillation (QBO), the 10.7 cm solar radio flux, El Niño Southern Oscillation (ENSO) without any lag applied, and the Aerosol Optical Depth (AOD, extended past 2012 by repeating the final available value from 2012 as the background AOD, which should be a valid assumption for Uccle). Four Fourier components representing the seasonal cycle are also included, unless (relative) monthly anomaly series are used as input ozone data. The output of the LOTUS MLR model and the different contributing terms (or proxies) for the monthly anomaly ozone concentrations at the layer 10 km above the tropopause (close to the ozone peak) are shown in Fig. S5. The final choice of those proxies (and possible lags) in LOTUS was based on retaining the optimal regression for global analysis of satellite data and broad latitude band analyses. Therefore, proxies describing rather local or small-scale phenomena might not have been included in the general "LOTUS regression" model. In particular, using an alternative stepwise multiple linear regression model for the Uccle stratospheric ozone amounts, we found that the Uccle tropopause pressure and the Arctic Oscillation are significant proxies as well (contributing statistically significant, i.e. at the 95% significance level of the *t*-test, to the regression coefficient). However, here, the analysis is limited to the LOTUS model and the sensitivity of the estimated trends on the chosen (M)LR model is rather limited for the Uccle time series.

The vertical profile of stratospheric ozone trends is shown in Fig. 2. From 1969 to 1997, stratospheric ozone concentrations decrease almost uniformly (and significantly) at a rate around -4 % dec$^{-1}$, except at the layers just above the tropopause.

Since 2000, the stratospheric ozone concentrations increase with about +2 % dec$^{-1}$, but only significantly at the layers below and at the ozone maximum (from 6 to 13 km above the tropopause, or 17 to 24 km for an average tropopause height of 11 km at Uccle). The insignificant negative trend of the Uccle ozone concentrations at the upper levels of Fig. 2 should be treated with caution, as the reliability of the ozonesonde instrument at those levels (above 30 km) is reduced. This is due to the increasing uncertainty in the pump efficiency at low pressures, the different stoichiometry of the chemical reaction due to a reduced amount of sensing solutions by evaporation, and frozen solutions. Additionally, an increase of the burst altitude in the Uccle ozonesonde time series in recent years and inhomogeneities due to changing pressure sensors with different radiosonde types might have an impact on the ozone trends at these very low pressures. In fact, the negative ozone trends are also less pronounced if calculated for absolute altitude levels. However, also for these altitudes, we prefer to calculate the vertical ozone trends in altitudes relative to the tropopause, to cancel out the seasonal variation of the ozone peak altitude, which roughly follows the tropopause height variation at Uccle: the ozone maximum peak is at its highest altitudes in summer (when the tropopause is also located higher), and is located at lower altitudes in winter (with the lowest tropopause). This approach gives in general vertical ozone trends that vary less over the different altitude levels. When we compare the post-2000 trends with those from the ozonesondes launched at De Bilt, the overall stratospheric positive insignificant trends apply for both stations, also at the higher altitude levels at De Bilt. The larger trend uncertainties for the De Bilt time series can be explained by the smaller frequency of launches (once a week versus three times a week at Uccle). The statistically insignificant offset between the Uccle and De Bilt trend estimates depends on the used correction methods at both sites, but also differences in the vertical ozone distribution (up to 5% in the stratosphere), of both geophysical and instrumental origin, have an impact on the trend values (see e.g. Figs. 10a and 12 in Van Malderen et al. (2016), in which a more detailed explanation of the differences in vertical ozone distribution and trends between Uccle and De Bilt is given).

The lower-stratospheric ozone trends deserve more discussion here, as Ball et al. (2018, 2019) reported a significant decline in lower stratospheric (13-24 km) ozone amounts for the periods 1998-2016 and 1998-2018 respectively, from multiple (merged) satellite measurements in the lower stratosphere between 60°N and 60°S. Also the latest Scientific Assessment of Ozone Depletion (WMO, 2018), largely based on the LOTUS final report (SPARC/IO3C/GAW, 2019), concluded that "there is some evidence for a decrease in lower stratospheric ozone from 2000 to 2016", although not statistically significant in most analyses. This decline, contradictory to the decline of ozone-depleting substances since 1997, is surprising and the current state-of-the-art chemical climate models (CCMs) used in Ball et al. (2020) and Dietmuller et al. (2021) do not show a decrease, but rather an increase of the lower-stratospheric mid-latitude ozone, although they confirm the lower-stratospheric ozone decline in the tropics in the observations. Using the Modern-Era Retrospective Analysis for Research and Applications Version 2 (MERRA-2) ozone output fields, Wargan at al. (2018) found a discernible negative trend of $-1.67 \pm 0.54$ Dobson units dec$^{-1}$ in the 10-km layer above the tropopause between 20°N and 60°N, and attributed the trend to changes driven by dynamical variations (as Chipperfield et al., 2018), in the form of enhanced isentropic mixing between the tropical (20°S–20°N) and extratropical lower stratosphere in the past two decades. In a follow-up study, Orbe et al. (2020), demonstrated that in the NH, this mid-latitude ozone decrease is primarily associated with changes in the advective

circulation rather than changes in mixing. In this study, both the Uccle and De Bilt time series do not show a significant decline in lower stratospheric (13-24 km) ozone amounts. On the contrary, although never significant, we found that the positive Uccle ozone trends in the lower stratosphere are rather robust, independent of the starting date (1997/1998/2000), 295   the used vertical coordinate system (absolute or relative to the tropopause), and the trend model used (LOTUS MLR or simple linear fit). The lower stratospheric ozone trends derived from the De Bilt time series show a larger variability between positive and negative statistically insignificant values, especially in the ten lowest kilometres.

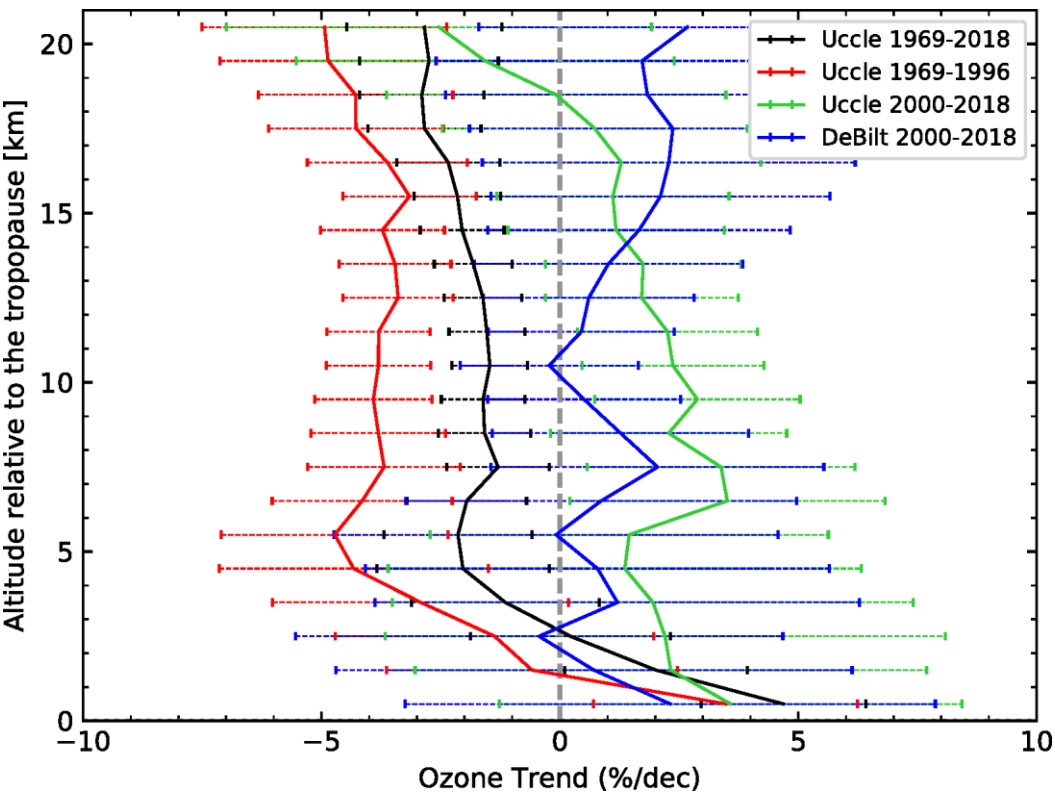

300   **Figure 2: Vertical distribution of trends of stratospheric ozone concentrations at Uccle for different periods (see text) and at De Bilt (2000-2018). The trends and their uncertainties are calculated with the LOTUS multiple linear regression model (see text and SPARC/IO3C/GAW, 2019), including an independent linear trend term. The 2-sigma error bars represent the trend uncertainty estimated by the regression model (using the fit residuals). For the Uccle 1969-2018 time series only, one linear trend term is included in the model instead. The output of the LOTUS MLR model and the different contributing terms for the monthly** 305   **anomaly ozone concentrations at the layer 10 km above the tropopause are shown in Fig. S5.**

Ball et al. (2020) investigated if the aforementioned changes in ozone and transport are also found in other stratospheric variables like the temperature. Globally, a reduction in lower stratospheric ozone should lead to reduced radiative heating and a decrease in observed temperature (see references in Ball et al., 2020). Quasi-global lower stratospheric temperatures from observations and in CCMs indeed decreased, with the post-2000 negative temperature trend being smaller compared to 310   pre-1998, mimiquing the observed lower-stratospheric ozone trends (Ball et al., 2020, but also Maycock et al., 2018),

although not the modelled ozone increase after 2000. On a smaller (European) scale, Philipona et al. (2018) found very similar seasonal and annual changes for temperature and ozone when averaging the Payerne, Hohenpeissenberg and Uccle ozonesonde measurements. With the exception of the fall season, annual and seasonal profiles switch from negative to positive trends before and after the turn of the century, for both ozone and temperature (see Fig. 4 in Philipona et al., 2018).

Here, on the local scale of Uccle and De Bilt, we also investigated the link between the lower-stratospheric ozone and temperature trends (see Fig. S6). Before 1997, the entire stratosphere above Uccle cooled significantly by -0.9 to -0.5 °C dec$^{-1}$, in line with the decreasing stratospheric ozone concentrations. After 2000, the stratospheric cooling at both Uccle and De Bilt ceased at the altitudes where ozone concentrations peak (see Fig. S6), and where their radiative impact on stratospheric temperatures is largest. Above and below the ozone maximum, the sign of the post-2000 temperature trends at Uccle

(respectively positive and negative) and De Bilt (respectively negative and positive) are reversed. As such, there is no direct imprint of the slightly positive lower-stratospheric ozone trends since 2000 in the temperature variability, in particular for Uccle. However, this might not be expected on a local scale, and in addition to ozone, stratospheric temperatures are affected by radiative effects from $CO_2$, $N_2O$, $CH_4$, as well as stratospheric water vapour, and chemical changes in these gases (Ball et al., 2020). These authors point to the increasing stratospheric water vapour amounts in the CCMs since 1996 in the mid-

latitudes, cooling the lower stratospheric, to reconcile the increasing lower-stratospheric ozone concentrations in the models with their stratospheric cooling over the same period and latitudes.

Finally, as we use the altitude relative to the tropopause as vertical coordinate, we should also mention the time variability of the tropopause height, which might impact the lower-stratospheric ozone trends. The tropopause height is increasing at both Uccle and De Bilt for all considered periods, but with different magnitudes: for Uccle, these are 6.98± 1.12 m dec$^{-1}$ (1969-

2018), 13.81± 3.00 m dec$^{-1}$ (1969-1996), and 11.62± 79.42 m dec$^{-1}$ (2000-2018), while for De Bilt the post-2000 trend magnitude is 25.73±19.23 m dec$^{-1}$. These increases in tropopause altitudes are consistent with results from the global study in Xian and Homeyer (2019) based on radiosondes and reanalyses, although with smaller magnitudes (they found increases of 40–120 m dec$^{-1}$ for the period 1981-2015). The thermal expansion of the troposphere and the associated increase in tropopause height have been proposed as robust fingerprints of anthropogenic climate change based on multiple

observational and model evidence (Santer et al., 2003, Seidel and Randel, 2006, Lorenz and DeWeaver, 2007).

We can conclude here that the Uccle stratospheric ozone trends before 1997 are well understood, but that the behaviour after 2000 is harder to explain, especially for the lower stratosphere, because of the lack of a clear link with stratospheric temperature variability and the impact of the tropopause variability. The link between the Uccle stratospheric ozone trends and these from the total ozone column measured with co-located spectrophotometers is discussed in Annex A.

**4.2  Tropospheric ozone trends**

Ozone in the troposphere is affected by many processes. Stratosphere-troposphere inflow and photochemical formation by interaction with sun light and ozone precursors ($NO_x$, CO and Volatile Organic Compounds) increase the ozone levels, while photochemical destruction of ozone in low $NO_x$ conditions (e.g. marine boundary layer and free troposphere, through OH-

HO$_2$ cycle) or at high NO$_x$ concentrations (urban regions under titration, i.e. via reaction with NO), and dry deposition on the ground removes ozone from the troposphere. Its short lifetime causes highly variable ozone concentrations in space and time, which complicates the understanding of the processes at play at all relevant spatio-temporal scales (Young et al., 2018). Moreover, the production of ozone in the troposphere is sensitive to variations in air temperature, radiation and other climatic factors (Monks et al., 2015).

Tropospheric ozone is measured with ozonesondes, by commercial aircraft, with different types of ground-based remote sensing instruments and with satellite instruments. Besides clear regional differences, the distribution and trends of ozone in the troposphere are not always consistent between these different datasets, and even not between different retrieval methods of the same satellite (e.g. Cooper et al., 2014, Gaudel et al., 2018). In fact, measuring the vertical profile of tropospheric ozone concentrations from satellites remains very challenging and relies on ground-based retrievals of ozone for validation (see Sect. 5).

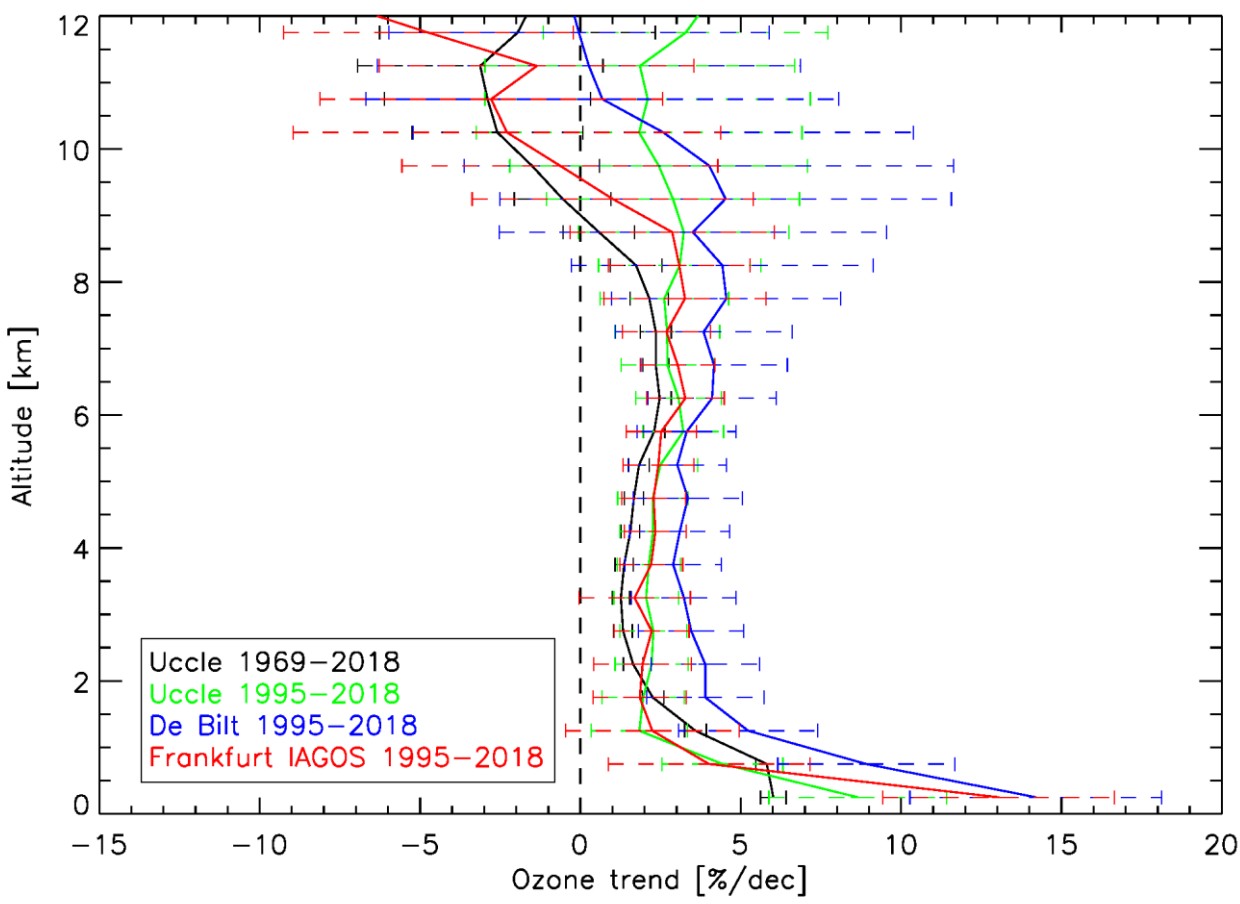

**Figure 3: Vertical distribution of trends of tropospheric ozone concentrations at Uccle for different periods, at De Bilt (ozonesonde data) and Frankfurt (IAGOS data) for 1995-2018. Simple linear trends are calculated for monthly ozone anomalies in 1 km altitude ranges and the error bars are the 2σ standard deviations. The same colour coding is used as in Fig. 2: trends for the most**

**recent Uccle sub-period (here 1995-2018) are in green, for the entire Uccle time series trends, we use black and the De Bilt trends are shown in blue. The red line denote the Frankfurt IAGOS trends in this figure.**

Here, we calculated the tropospheric ozone trends from the Uccle and De Bilt ozonesonde time series and the MOZAIC (Measurement of Ozone and Water Vapour by Airbus in-service Aircraft) and IAGOS (In-service Aircraft for a Global Observing System) ascent and descent profiles at Frankfurt airport, at about 320 km from Uccle. This MOZAIC-IAGOS dataset consists of more than 27600 profiles, starting in August 1994, and is combined with the data from Munich airport,

approximately 300 km southeast of Frankfurt, between 2002 and 2005 (about 4200 flights), to fill a large data gap in 2005 (also done in e.g. Petetin et al., 2016). With typical horizontal ozone correlation lengths of about 500 km in the troposphere (Liu et al., 2013), some correlation of especially free-tropospheric ozone trends between Uccle on one hand, and De Bilt and Frankfurt on the other hand, is expected. We used simple linear trends based on the monthly anomalies at different altitude levels (see Fig. 3), as there is no consensus in using (which) proxies to account for natural variability. First, for the 1995-

2018 period, the extremely good agreement between the Uccle (in green in Fig. 3) and IAGOS (in red) vertical ozone trends in the free troposphere (3-8 km) is striking. Although the integrated tropospheric ozone amounts for this altitude range are lower for the region above Frankfurt (14.9 DU) than above Uccle (16.2 DU), the overall relative trends are similar (resp. 2.09 ±1.01 % dec$^{-1}$ and 2.47 ±1.01 % dec$^{-1}$, see Fig. S7). The De Bilt trends (in blue in Fig. 3) are larger in the free troposphere, with also larger uncertainties, due to the lower launch frequency. In this context, we mention the sensitivity

analysis of IAGOS profiles above Europe by Chang et al. (2020), which concluded that an optimal sample frequency of 14 profiles per month is required to calculate trends with their integrated fit method (and about 18 profiles a month when this method is not used). Near the surface, the De Bilt trend is in better agreement with the Frankfurt trend, but the local surface ozone production and destruction and the boundary layer dynamics can vary substantially between the three sites considered here, so that the boundary ozone distribution and trends at the three sites are likely to be uncorrelated. However, comparing

the lower-tropospheric IAGOS measurements at Frankfurt with nearby (within 50-80 km) and more distant (within 500 km) surface stations, Petetin et al. (2018) showed that the IAGOS observations in the first few hundred meters above the surface at Frankfurt airport have a representativeness typical of suburban background stations (like e.g. Uccle and De Bilt), and as one moves higher in altitude, the IAGOS observations shift towards a regional representativeness. A detailed description of the surface ozone trend at Uccle and its relation with ozone precursor trends is provided in Appendix B.

In the upper troposphere, the ozone concentration trends deviate more between the different datasets, both in magnitude and sign, with larger trend uncertainties. At these altitudes, the aircraft could be very distant from Frankfurt (or Munich) airport, as the ascent/descent profiles stop/start at about 400 to 500 km from the airport. The measurements at these altitudes are hence representing large areas. Therefore, the closer agreement between the Uccle and De Bilt trends above 8 km compared to the IAGOS trend might be attributed as well to a similar source region. Also, at those altitudes, the trends do not represent

the tropospheric ozone temporal variability only, as the mean tropopause height range between 10.5 km (winter time) and 11.5 km (summer time), with standard deviations between 1 and 1.5 km, both at Uccle and De Bilt. As a consequence, lower-stratospheric ozone concentrations will contribute to the estimated trends in the upper altitude levels of Fig. 3.

The Uccle tropospheric ozone concentrations have been increasing at about the same rate since 1969 (in black in Fig. 3) as since 1995 (in green in Fig. 3), and also the post-2000 increase rate is very similar (not shown here, but is also suggested in

the tropospheric ozone column time series shown in Fig. S7). The increase in (free) tropospheric ozone concentrations above Uccle until the early 2000s is consistent with the findings reported above (Western) Europe in the literature review of Cooper et al. (2014). Over the 2000-2014 period, the emissions of the key ozone precursor, nitrogen oxides ($NO_x$), declined in North America and Europe due to transportation and energy transformation (Hoesly et al., 2018). Therefore, the overall increase in ozone concentrations has flattened, but resulted in spatially and seasonally varying tropospheric ozone trends

over North America and Europe, without consistency in even the sign of the ozone trends (Gaudel et al., 2018, and references therein). However, Cooper et al. (2020) concluded, based on the IAGOS observations, that the Western Europe free-tropospheric trends since 1995 are predominantly positive. Using a different statistical approach, i.e. a nonlinear regression fit of a quadratic polynomial to normalized, deseasonalized monthly mean ozonesonde (merged data from Uccle, Hohenpeissenberg, and Payerne) and MOZAIC/IAGOS data (Frankfurt) between 3 to 4 km altitude, Parrish et al. (2020)

indicated that those ozone concentrations increased through the 1990s, reached a maximum in the years 2001 (merged ozonesonde) and 2007 (IAGOS) and have since then decreased.

To explain the tropospheric ozone concentration trends, Griffiths et al. (2020) used a chemistry-climate model employing a stratosphere-troposphere chemistry scheme, and found that for the period 1994-2010, despite a levelling off in emissions, increased stratosphere-to-troposphere transport of recovering stratospheric ozone drives a small increase in the tropospheric

ozone burden. Taking advantage of the high vertical resolution of the ozone profiles and the high frequency of launches at Uccle, we focus on the time variability of specific cases of deep intrusions of stratospheric air into the troposphere, i.e. tropopause folds. Akritidis et al. (2019) stressed the role of tropopause folding in stratosphere-to-troposphere transport (STT) processes under a changing climate, suggesting that tropopause folds will be associated with both the degree of and interannual variability in ozone STT. Tropopause folds occur because of the ageostrophic circulation at the jet entrance and

coincide with the frontal zone beneath the jet. The automatic algorithm applied in this work detects tropopause folds in the Uccle ozone sounding profiles as ozone rich (two criteria), stable (one criterion) and dry (one criterion) air mass layers located in an upper level front in the vicinity of an upper tropospheric jet stream (two criteria), and is described in Van Haver et al. (1996). This identification by means of those six criteria is also illustrated in an example of an ozone sounding containing a tropopause fold in Fig. S8.

Tropopause folds are rather rare events at Uccle: out of the 6526 soundings analysed for the 50 year period (1969-2018), only 290 soundings (or 4.4%) showed evidence of a tropopause folding. However, similar occurrence rates (between 3 and 10%) have been found over Europe at French ozonesonde sites (Beekmann et al., 1997) and with other techniques (Rao et al., 2008, and Antonescu et al., 2013). On a monthly scale, most folding events occur in March, June, July and August (occurrence > 5%), whereas in January, April, May and December, the amount is lower (Fig. 4). What is most important here

within the context of the tropospheric ozone trends is the dramatic increase of the amount of tropopause folding events over time with $0.14 \pm 0.02\%$ yr$^{-1}$ (see also Fig. 4). Van Haver et al. (1996) detected a smaller and insignificant trend of $0.07 \pm$

0.11% yr⁻¹ at Uccle for the 1969-1994 period. On one hand, the large increase over the entire time period might be explained due to some technical aspects. First, the higher vertical resolution of the sounding data in the more recent digital era (since 1990) might have an impact on the larger detected number of tropopause folds (thinner layers might be detected), although the amount of events has continuously increased since then, at a slightly smaller annual rate of $0.12 \pm 0.05\%$. Secondly, a visual inspection of all profiles fulfilling at least five of the tropopause fold detection criteria, led to a higher number of (manually) identified events (around 50 more), and (relatively) especially in the beginning of the time series. This is explained by the fact that the low humidity criterion was often not met in the automatic detection, because there were no humidity data or the humidity sensor was iced (following the icing recognition algorithm of Leiterer et al., 2005). More recent types of radiosonde humidity sensors (in use since 2007 at Uccle) prevent ice contamination by heating them during flight. However, this manual (and hence more subjective) mode of the algorithm still gives a $0.09 \pm 0.02\%$ annual increase of the tropopause fold events since 1969. Therefore, we believe that the significant increase, although possibly overestimated by the automatic procedure, is nevertheless a robust feature of the analysis. Additionally, a higher rate of tropopause folding events is expected due to climate change (Tarasick et al., 2019, and references therein): climate change is projected to increase planetary wave activity inducing an accelerated Brewer-Dobson circulation. This acceleration, along with stratospheric ozone recovery, will lead to increased transport of ozone from the stratosphere into the troposphere and hence more tropopause folding events. Akritidis et al. (2019) elaborated that the degree of increase in the downward transport of stratospheric ozone is partially driven by the long-term changes in tropopause fold activity.

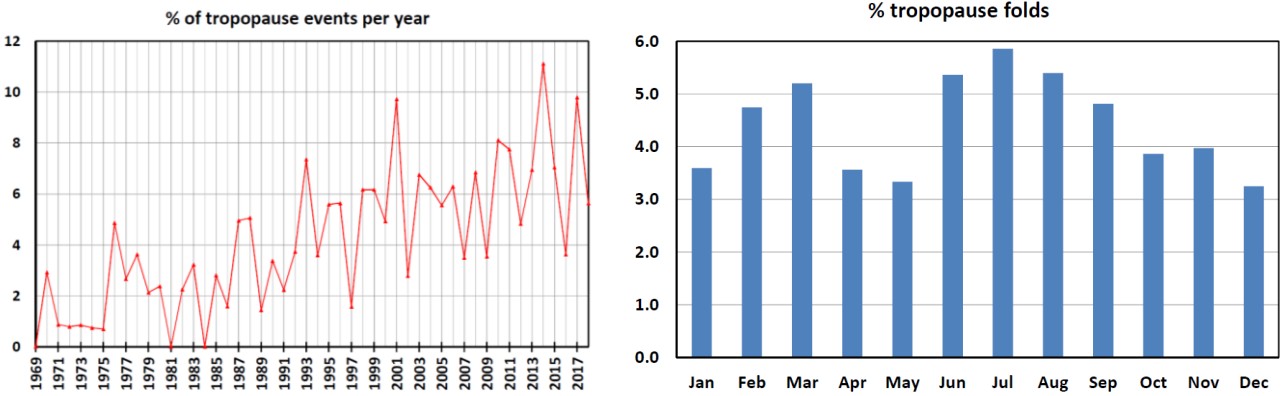

**Figure 4: Left: Relative frequency of detected tropopause folding events per year in the ozone soundings at Uccle. Right: Relative frequency of tropopause folding events per month.**

To conclude, we found very consistent positive vertical tropospheric ozone trends between Uccle, De Bilt, and Frankfurt (IAGOS) since 1995, which are consistent with other studies, both observational and from a modelling approach, but different processing and statistical methodologies can result in different European trend patterns for the last two decades.

## 5    Validation of satellite ozone retrievals with Uccle ozonesonde data

Ozonesondes are virtually all-weather, i.e., unaffected by clouds and precipitation, in contrast to most spectroscopic techniques, and they provide high vertical resolution ozone profiles from the ground to about 30 km. Therefore, satellite algorithms are based on ozonesonde climatologies and in turn satellites are validated by the sondes. Since the start of the ozone measuring satellite era, ozone profiles from soundings at Uccle have been used for validation of satellite ozone retrievals, e.g. the Stratospheric Aerosol and Gas Experiment (SAGE) II satellite profiles (Attmannspacher et al., 1989, De Muer et al., 1990). In this section, we give some recent examples of the application of the Uccle ozone profile data for operational satellite validation (Sect. 5.1), and for the scientific evaluation of both stratospheric (Sect. 5.2) and tropospheric (Sect. 5.3) ozone profile retrievals by satellite instruments. In these latter two sections, we also illustrate that a consistent and homogenous ozonesonde dataset like the Uccle one is crucial to determine the long-term stability of (merged) satellite ozone retrievals.

### 5.1  Operational validation within EUMETSAT AC-SAF

As partner of the EUMETSAT Atmospheric Composition Satellite Application Facilities (AC SAF), RMI is responsible for the validation of different ozone products (ozone profiles and (tropical) tropospheric ozone columns, see Hassinen et al., 2016, Valks et al., 2014, van Peet et al., 2014) from Global Ozone Monitoring Experiment GOME-2 and Infrared Atmospheric Sounder Interferometer (IASI) instruments on board the MetOp A/B/C satellite platforms. Those different instruments give the opportunity to obtain a unique dataset, retrieved with an identical technique, from the beginning of the MetOp-A/GOME-2 instrument in 2007, until the end of the third, MetOp-C/GOME-2, foreseen in 2022. GOME-2 ozone profiles are given as partial ozone columns, expressed in Dobson Units, on 40 varying pressure levels between the surface level and 0.001 hPa and are calculated by the Ozone Profile Retrieval Algorithm (OPERA, van Peet et al., 2014). The a-priori information used for the retrieval is obtained from McPeters and Labow (2012).

For the validation of GOME-2 ozone profiles within the AC-SAF, ozonesonde measurements are extensively used. However, for a meaningful comparison, the ozonesonde profiles need to be integrated first between the GOME-2 pressure levels. When comparing a single ozonesonde profile with different GOME-2 profiles, the actual reference ozone values are not identical due to the fact that the GOME-2 vertical levels vary from one measurement to another. GOME-2 has a nominal spatial resolution of 80 km x 40 km, but for the shortest UV wavelengths the integration time takes eight times longer because of the lower number of photons arriving on the detector pixels. Secondly, as the ozonesondes and the satellite do not have the same vertical resolution, it is necessary to take into account the averaging kernels (AVK), to "smooth" the ozone soundings towards the resolution of the satellite (Rodgers, 2000).

In Fig. 5 the relative differences between the MetOp-A operational ozone profile product and the Uccle ozonesonde profiles are shown for the year 2018 (red colour). The following co-location criteria were applied: a geographic distance of less than 100 km between the GOME-2 pixel centre and the sounding station location, and a time difference of less than 10 hours

between the pixel sensing and the sounding launch time. The figure highlights two different aspects of the operational validation. First, it can be noted that applying the averaging kernels to the sounding profiles improves the comparison with the GOME-2 ozone product significantly, i.e. by 15%, in particular in the lower stratosphere (compare the full lines with dashed lines in Fig. 5). The lower stratosphere is the region with the highest ozone variability, so smoothing the high resolution ozonesonde profiles to the GOME-2 vertical resolution will have the largest effect here by removing details of the differences. Secondly, as the GOME-2 ozone profile product is based on UV measurements, it is sensitive to degradation of the UV sensor (van Peet et al., 2014, Munro et al., 2016). For example, the measured values of the GOME-2A irradiance in the UV (below 300nm) has reduced by roughly 80% in 2016 (since its launch in 2007). Since the vertical ozone profile retrieval algorithm depends on an absolute calibrated reflectance (sun normalised radiance) there is a need to correct for this temporal change of the (joint) radiance and irradiance. This method depends on the assumption that, taken as an average across the globe, the atmospheric constituents (mainly ozone) will be close to the multi year climatological value from McPeters and Labow (2012). The climatological ozone profile is then scaled with the Assimilated Total Ozone columns to get the overall ozone absorption correct (Tuinder et al., 2019). This degradation correction has been applied to the data for the relative differences with the Uccle data in Fig. 5 (in grey). From this figure, it should be clear that this degradation correction improves significantly the agreement with the Uccle ozonesonde data compared to the operational product (in red), resulting in relative differences between GOME-2 ozone profiles and the Uccle data within the target error range of 15% (marked by the vertical red lines). The improvement after degradation correction is a promising result, showing the challenge for UV-VIS sounders to obtain a stable ozone profile product on different sensors (GOME-2A/2B/2C) for different periods using the same type of optical instrument. More feedback on the status of the operational EUMETSAT product can be obtained in the validation reports, available on the AC SAF website (https://acsaf.org, e.g. Delcloo and Kreher, 2013).

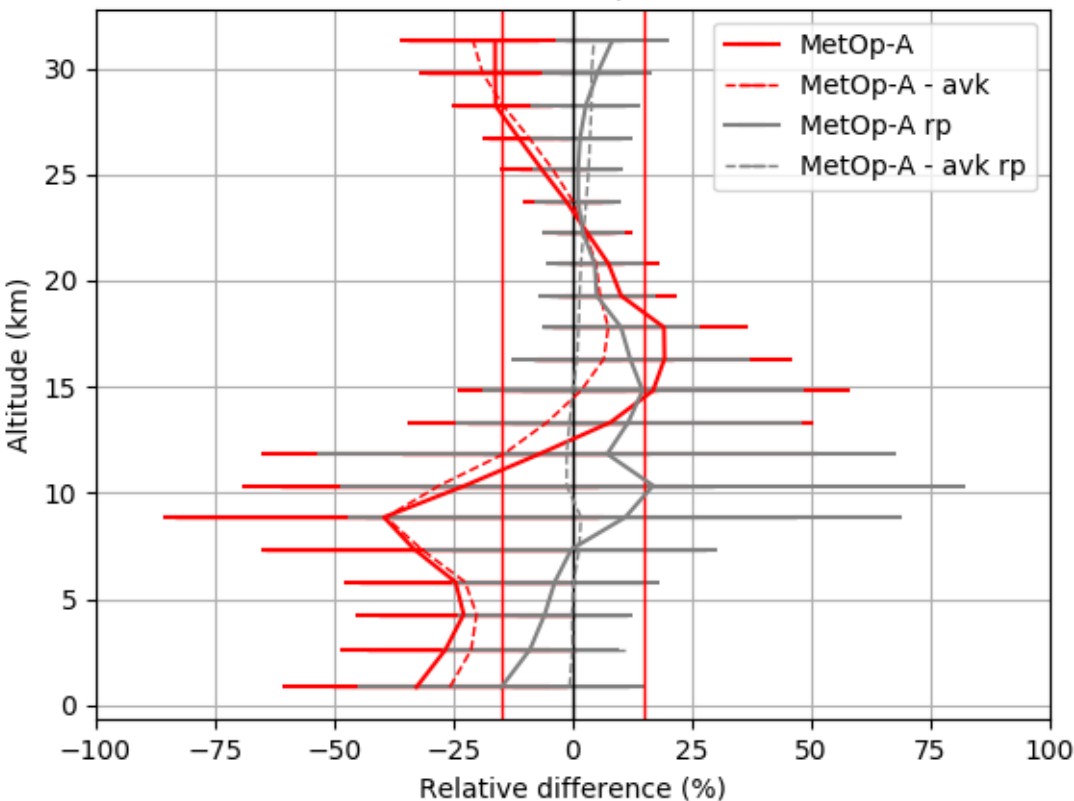

**Figure 5: Relative mean differences and standard deviations between ozone profiles retrieved from MetOp-A/GOME-2 and Uccle ozone profiles for the time period January to December 2018. The red graph represents the mean differences when using the operational MetOp-A/GOME-2 product, the grey graph when the UV sensor degradation correction has been applied in the MetOp-A/GOME-2 ozone retrieval. Relative mean differences denoted by dashed lines are obtained after applying the averaging kernel to the Uccle sounding data. Finally, the thin red vertical lines mark the ±15% target error range of the MetOp-A/GOME-2 ozone profile product.**

## 5.2 Validation of AURA-MLS stratospheric ozone profiles

The Microwave Limb Sounder (MLS, Froidevaux et al., 2008) is one of the four instruments on the Earth Observing System (EOS) Aura Satellite. MLS has been measuring vertical profiles of atmospheric trace gases, including ozone, along with temperature, geopotential height, relative humidity, cloud ice water content and cloud ice water path, since its launch in 2004. Global measurements (from 82° S to 82° N), in a near-polar, sun-synchronous orbit, at two fixed solar times, noon-night, at around 01:30 a.m./p.m. are achieved, with the number of profiles over e.g. ozonesonde sites varying between 0 and 6 daily. MLS products have been validated to be very accurate and stable (Jiang et al., 2007, Froidevaux et al., 2008), and have been used in many studies involving ozonesonde measurements (e.g. Witte et al., 2017, Stauffer et al., 2020). Here, we

have implemented the MLS v4.2 data, screened according to the v4.2 Level 2 MLS Data Quality document (Livesey et al., 2020), and compared the satellite overpass measurements with coincident ozonesonde profiles at Uccle. Because there are multiple profiles crossing over Uccle at a fixed time, the profile closest in distance is used for the validation. Both the noon and night overpasses have been used, as we did not find significant differences between those. As a result, ~3000 profiles were included into the validation. Thanks to the relatively dense and regular MLS vertical resolution of around 2.5 km in the 10-200 hPa pressure range, it is feasible to interpolate the Uccle ozonesonde data to the MLS pressure levels on a fine pressure grid of 2.5 km. Applying the time invariant MLS averaging kernel on the latitude of Uccle on the ozonesonde data did not have a large effect on the smoothing of the vertical ozonesonde profile, as compared to applying the identify matrix to the ozonesonde vertical profile (< 1%). This contrasts strongly with the GOME-2 and TES retrievals (see Sect. 5.3), where the spatio-temporal varying averaging kernels affect the vertical ozone profiles substantially, and as such should be used on the sonde data for pairwise comparison. The mean annual relative differences between MLS and Uccle ozonesondes are shown in Fig. 6. Different conclusions can be drawn from this figure. First, MLS and the Uccle ozonesondes compare very well, within ±5% between 10 and 70 hPa (grey shading in Fig. 6). At pressures smaller than 10 hPa, ozonesonde measurements are systematically underestimating ozone due to the evaporation or freezing of the sensing solutions (see also the composite ECC-MLS Fig. 3 in Stauffer et al., 2020), and they have a larger uncertainty due to increased pump efficiency uncertainty at low pressures. On the other hand, at pressures larger than 70 hPa, the MLS ozone retrieval is more challenging because of the longer atmospheric path and the lower ozone volume mixing ratios increasing the relative differences. Another important finding from this figure is that the mean annual relative differences are very consistent over the different years, which means that both the MLS instrument and the Uccle ozonesonde time series are very stable with respect to each other. In addition to this, we also want to mention that the relative differences between MLS and Uccle ozonesondes are very similar for the different seasons, see Fig. S9.

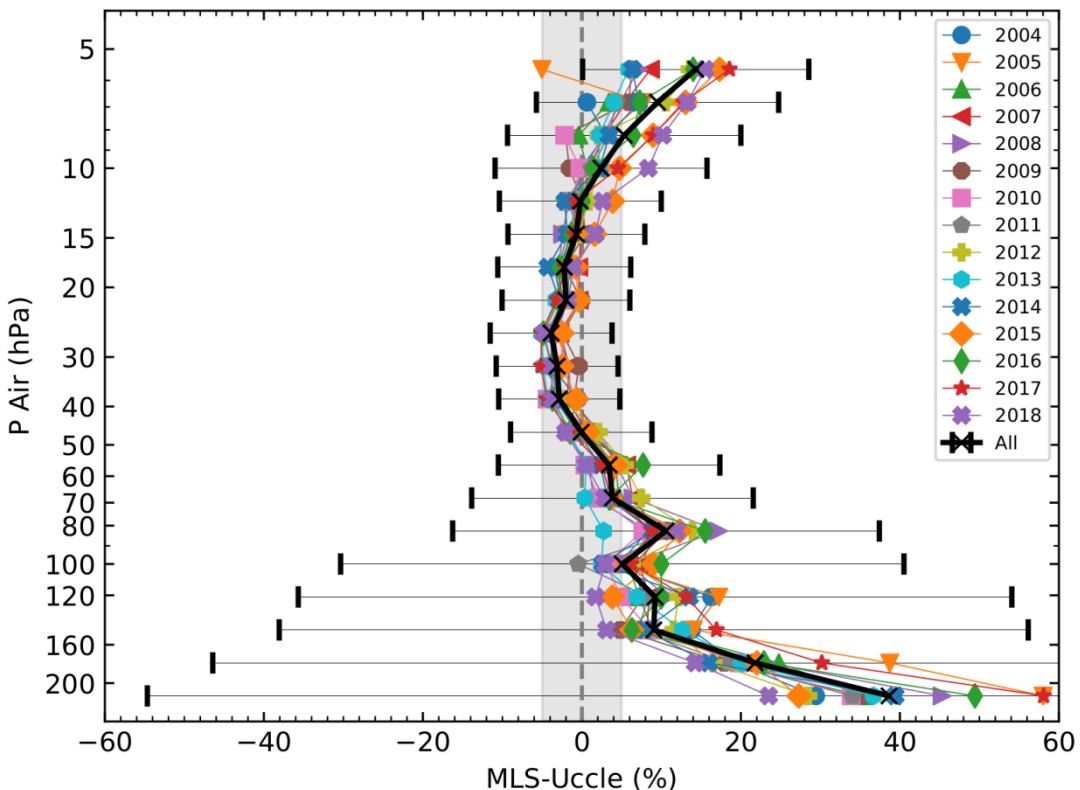

**Figure 6: Relative ozone profile differences between MLS and Uccle ozonesondes. The different colours correspond to the different yearly averages, illustrating the large consistency among those. The black line represents the overall mean relative difference, with the error bars the one standard deviations due to the individual differences. Note that individual differences are relatively large at some pressure levels, but they are cancelled out in the yearly mean .**

### 5.3 Validation of AURA-TES tropospheric ozone profiles

Here we compare the tropospheric vertical ozone profiles of the Uccle sondes coinciding with the observations from the Tropospheric Emission Spectrometer (TES) sensor on-board the Aura satellite for the period late 2004 to early 2018, when the instrument was decommissioned. TES is an infrared Fourier transform spectrometer (Beer et al., 2001; Beer, 2006) that measures radiance spectra of Earth's atmosphere, predominantly nadir viewing, at wavelengths between 3.3 and 15.4 µm. The nadir vertical profiles are spaced 1.6° apart along the orbit track and have a footprint of approximately 5×8 km² (Beer et al., 2001; Beer, 2006).

The vertical sensitivity of the TES-retrieved ozone is the largest for the troposphere, with a vertical resolution for ozone profiles of 6-7 km, corresponding to 1-2 degrees of freedom in the troposphere (Jourdain et al., 2007). Prior to applying TES

ozone data, they are subject to screening, using the TES ozone master quality flag that accounts for clouds and a too large difference between observed and simulated radiances (Osterman et al., 2008).

As in Nasser et al. (2008) and Verstraeten et al. (2013), we apply temporal and spatial coincidence criteria of ±9 h and ±300 km respectively between the sonde and TES observations. These criteria can provide enough profiles for a statistically meaningful comparison while it is sufficiently strict to warrant a high probability that both instruments sample similar air masses. A mapping matrix is used to interpolate the sonde data to the 67-level pressure grid (from 1212 to 0.1 hPa) used in the TES retrievals. Then, the TES averaging kernel was applied to the 67-level pressure grid of the Uccle sonde data to

ensure a consistent comparison between TES and ozonesonde data excluding the influence of the a priori ozone profile needed to regulate the TES retrieval (Verstraeten et al., 2013).

By applying all these constraints (coinciding criteria and the TES ozone master flag), 191 suitable coincidences or data pairs for the full time range from 2004 to 2018 were collected. The upper panels in Fig. 7 present TES−sonde tropospheric ozone profile differences for the Uccle sondes. The left upper panel shows the absolute ozone vertical profile differences (TES−

sonde) in the troposphere (1000–300 hPa). The right upper panel shows the relative differences ((TES−sonde)×100/sonde) for the full vertical ozone profile (1000– 1 hPa).

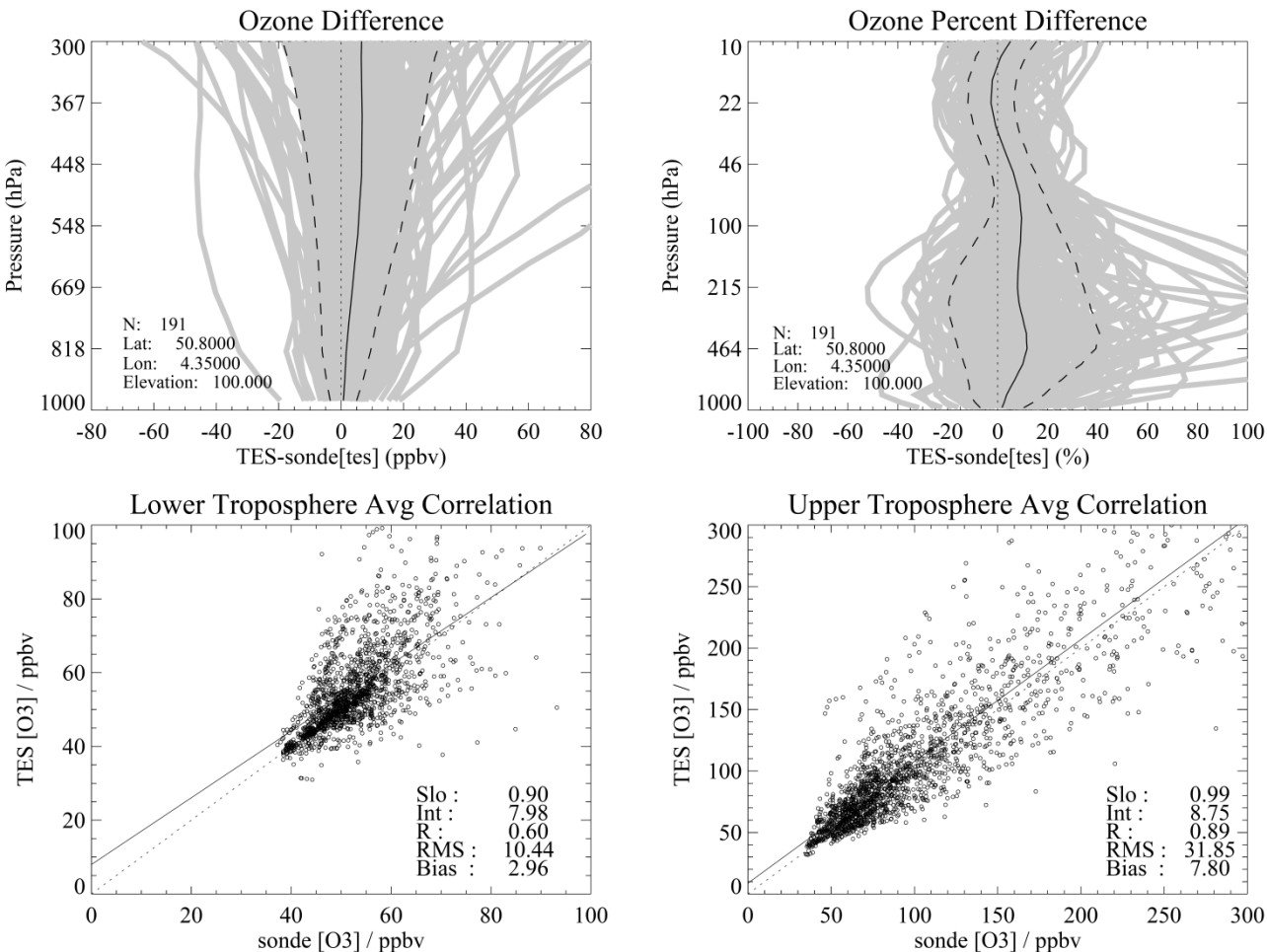

**Figure 7: Absolute TES–sonde tropospheric ozone vertical differences (left upper panel) and relative differences (right upper panel) for the whole profile of Uccle. Individual difference profiles are shown in grey; the mean difference and one standard deviation profiles are in black. N is the number of valid profiles after flagging TES data and using the maximum 300 km and 9h coinciding criteria. Lower left panel illustrates the correlation between TES and sonde O₃ for the lower troposphere (1000 to 500 hPa) including the slope (slo), intercept (Int) and correlation (R) of the linear regression, the root mean square error (RMS) and the bias. Similar for the lower right panel for the upper troposphere (500 hPa to tropopause).**

Figure 7 indicates that TES is generally positively biased within the troposphere by up to ~10 ppbv, corresponding to relative

differences up to ~15 %. The TES bias slightly varies as a function of pressure. TES appears to be almost unbiased with

respect to the sondes in the lower troposphere, but this actually reflects the non-sensitivity of TES to ozone in the lower

atmosphere for situations with lower brightness temperature as encountered at higher latitudes. Since the TES signal in the

troposphere has typically 1–2 degrees of freedom, analysing the TES bias for two vertical regimes - the lower troposphere

(LT, 1000 to 500 hPa) and the upper troposphere (UT, 500 hPa to tropopause)- might be meaningful (Nassar et al., 2008).

From a linear regression of all TES vs. sonde ozone data pairs for Uccle in the lower troposphere (Fig. 7, lower left panel),

we find a slope of 0.90 with an intercept of 7.98 (R = 0.60) with a bias of +2.96 ppbv. For the upper troposphere (lower right

panel in Fig. 7) the bias is a bit higher (7.80 ppbv), the correlation (R) is 0.89 and the slope and intercept are 0.99 and 8.75 respectively. These values are in line with reported ones for data pairs collected for the whole northern mid-latitudes (Verstraeten et al., 2013).

The temporal stability of the TES sensor for tropospheric ozone can be assessed by applying the Theil Sen trend statistics (Theil 1950a, 1950b, 1950c; Sen, 1968) on the time series of the TES-sonde data pairs for each pressure level in the troposphere (surface to 300 hPa). Analysis shows that all p-values are larger than 0.05 indicating that all slopes of the linear regression are not statistically different from zero in the troposphere. All $R^2$ values are smaller than 0.01. Thus, there is no reason to assume any temporal trend for data pairs in the troposphere. This is in line with the same analysis for the 464 hPa

level by Verstraeten et al. (2013).

## 6    Conclusions and outlook

Having started operationally in 1969 to use ozone as a tracer to study the general air circulation in the troposphere and the lower stratosphere, the high-frequency (three times a week) mid-latitude Uccle ozone sounding time series now extends over more than 50 years, covering over 7000 profiles. Over this entire period, attention has always been paid to the consistency of

the time series, resulting in only one major change: the switch from BM to En-Sci ECC sondes in 1997. This change was however well documented with dual launches and pump efficiency laboratory measurements of both pump types, so that a unique correction method for both sonde types, a PRESsure and Temperature dependent total Ozone normalization (PRESTO, Van Malderen et al., 2016), has been developed (De Backer et al., 1998a,b) to guarantee the data homogeneity. Another distinct feature of the Uccle ozonesonde dataset is the correction for urban $SO_2$ interference with the chemical

reactions in the ozone cells in the first half of the period.

Although satellites provide global routine measurements of ozone profiles with increasing accuracy and spatial resolution, ozonesondes are the only technique that can provide, since 50 years, accurate (around 5-10%), vertically resolved observations from the surface up to the lower stratosphere, unaffected by clouds or precipitation. Furthermore, they can resolve strong gradients in the UTLS (upper troposphere/lower stratosphere), while precisely locating the thermal tropopause

(Thompson et al., 2011). In this paper, we illustrated the importance of the Uccle ozonesonde dataset in two specific application areas: for the assessment of the long-term vertical ozone trends and for the validation of satellite retrievals of ozone profiles. The strength of the ozonesonde measurements (and the Uccle time series in particular) lies exactly in combining those two aspects of ozone research, together with its applicability in process studies. The major conclusions are summarized here.

Making use of the LOTUS multiple linear regression model including the QBO, the solar radio flux, ENSO, and AOD as explanatory variables, we found that the stratospheric ozone concentrations at Uccle declined at a significant rate of around 2% dec$^{-1}$ since 1969. This overall decline can mainly be attributed to the increasing ODS emissions, with a rather consistent decline rate around -4% dec$^{-1}$ for the period 1969-1996. Since 2000, a recovery between +1-3% dec$^{-1}$ of the stratospheric

ozone levels above Uccle is observed, although not significant and not for the upper stratospheric levels measured by ozonesondes. A significant decline in lower stratospheric ozone amounts since 1998, as reported by Ball et al. (2018, 2019), is hence not present in the Uccle and nearby De Bilt time series. For the considered periods, we found an overall agreement between the sign of the stratospheric temperature trends and those ozone concentration trends, i.e. a cooling of the stratosphere in 1969-2018 and 1969-1996 and an insignificant warming for all but the lower stratospheric layers since 2000, underlining the possible mutual interaction between stratospheric ozone concentration and temperature changes.

In Appendix A, we showed that the total column ozone loss at Uccle between 1971-1996 (at a rate of -1.6% dec$^{-1}$) has nearly fully recovered by the +1.9% dec$^{-1}$ gain since 2000. In the light of the discussion on the stratospheric ozone trends in the previous paragraph, this would mean that the tropospheric ozone amounts at Uccle should increase since the mid-90s. We indeed confirmed a very consistent increase of the ozone concentrations at 2 to 3 % dec$^{-1}$ throughout the entire free troposphere, a number which is in almost perfect agreement with the trends derived from the IAGOS ascent/descent profiles at Frankfurt, and 1% dec$^{-1}$ lower than the De Bilt tropospheric ozone trends. The Uccle 1995-2019 trend is even 0.5 to 1% dec$^{-1}$ higher than the 1969-2019 trend. Despite the levelling off in tropospheric ozone precursor emissions, the tropospheric ozone amounts in Uccle are still increasing. Based on chemistry-climate model calculations, Griffiths et al. (2020) found that an increase in the tropospheric ozone burden might be driven by increased stratosphere-to-troposphere transport of recovering stratospheric ozone. It should also be noted that the amount of tropopause folding events in the Uccle time series increased significantly over time, which might be an indicator for increased transport of ozone from the stratosphere into the troposphere. However, in line with the free-tropospheric ozone, the surface ozone concentrations at Uccle continue to increase since the beginning of those measurements in the 1980s, despite the decreasing on-site concentrations of precursor trace gases CO, NO, and $NO_2$ (see Appendix B).

For the operational validation of the GOME-2 and IASI ozone profiles within the EUMETSAT AC-SAF, the role of ozonesonde profiles is crucial. We showed how the Uccle dataset can be used to evaluate the performance of a degradation correction for the GOME-2 UV sensors. The Uccle ozonesondes are also used to assess the accuracy and stability of satellite ozone retrievals. Here, we showed that the AURA-MLS overpass ozone profiles agree very well with the ozonesonde profiles, within ±5% between 10 and 70 hPa. Another instrument on the same AURA satellite platform, TES, has its largest vertical sensitivity for ozone in the troposphere, and is generally positively biased with respect to the Uccle ozonesondes in the troposphere by up to ~10 ppbv, corresponding to relative differences up to ~15 %. Using the Uccle ozonesonde data series as reference, we also found that the temporal stability of both satellite retrievals is excellent. Vice versa, satellite total ozone retrievals and MLS have enabled the detection of a post-2013 drop-off in total ozone at a third of global ozonesonde stations (Stauffer et al., 2020), a number now reduced to about 20% (12 of 60 global stations, Stauffer et al., 2021, private communication). Our analysis with MLS here confirmed their finding that Uccle is not affected by any total column drop-off of more than 3% in its time series.

 A higher flexibility of ozonesonde launch times toward satellite overpass times is an emerging issue that needs to be considered against the preference for a fixed launch time for e.g. the assessment of tropospheric ozone trends. Moreover, for

over a decade, weather prediction centres have been incorporating chemistry into operational forecasts, assimilating satellite ozone retrievals, and ozonesondes are used for external evaluation of those model forecasts (e.g. for tropospheric ozone:

Flemming et al., 2015), analyses (e.g. for stratospheric ozone: Lefever et al., 2015) and reanalyses (e.g. Inness et al, 2019). Those services require a near real-time delivery of the ozonesonde measurements, with an operational quality assessment/quality control tool. These are the challenges for operational applications of ozonesondes. For the assessment of the long-term variability of ozone concentrations at different atmospheric altitudes and the interaction between climate change and ozone (also studied in coupled chemistry-climate and chemistry-transport models, see e.g. Morgenstern et al.,

2017), the availability of a long-term homogeneous dataset is crucial. Homogenization efforts of ozonesonde networks and/or datasets (Tarasick et al., 2016; Van Malderen et al., 2016; Thompson et al., 2017; Witte et al., 2017, 2018, 2019; Sterling et al., 2018) should therefore be continued and extended. With these developments in mind, we aim at continuing the pioneering role that the Uccle time series had in some of the research areas during its half a century lifetime.

## Appendix A: The Uccle total ozone trends

The total column ozone amounts at Uccle, available since 1971, are retrieved with a Dobson UV-spectrophotometer (no. 40, 1971-1989), a single Brewer UV spectrophotometer (no. 16, 1990-current, but used in the time series until the end of 2001), and a double Brewer UV spectrophotometer (no. 178, 2002-current). The calibration history of the Dobson instrument is documented in De Muer and De Backer (1992) and the transition to the Brewer instrument is described in De Backer and De Muer (1991). Both Brewer instruments were recalibrated against the traveling standard Brewer instrument no. 17 in 1994

(no. 16 only), 2003, 2006, 2008, and against the travelling reference Brewer no. 158 since 2010 every second year. The stability of the instruments is also continuously checked against the co-located instruments (with the Dobson no. 40 from 1991 until May 2009, between both Brewers since 2001). Internal lamp tests are performed on a regular basis to check whether a Brewer instrument is drifting. When instrumental drift is detected, it is corrected for.

The time series of total ozone measurements is shown in Fig. A1, but has been smoothed by applying a low-pass Gaussian

filter with a width at half height of 12 months, to filter out variations with frequencies higher than one year. With this representation, the impact of the major (strato)volcanic eruptions of Fuego (Guatemala, Oct 1974), El Chichon (Mexico, Mar/Apr 1982), and Pinatubo (the Philippines, Jun 1991) is shown in the significant dips in Uccle total ozone. Indeed, the episodes of enhanced stratospheric aerosol-related ozone loss after those major volcanic eruptions are confirmed by model results (see e.g. Tie and Brasseur, 1995, Solomon 1999, Aquila et al., 2013 for a description of the mechanism behind) and

can clearly be identified in the time series. Also the other inter-annual variability in Fig. A1 is very similar to the Northern Hemisphere (NH) annual mean total ozone time series of five bias corrected merged datasets in the 35–60° N latitude band in Weber et al. (2018; their Fig. 2). In 2010, the Uccle ozone levels were unusually high, as over the entire NH extratropics. An unusually pronounced and persistent negative phase of the Arctic Oscillation and North Atlantic Oscillation in 2010, with the co-incidence of northern winter 2009/2010 with the easterly wind-shear phase of the QBO have been identified as major

contributors (Steinbrecht et al., 2011) of this excess ozone. The 2011 ozone low anomaly cannot be fully explained by including this Arctic Oscillation and other dynamical proxies (e.g. for the Brewer-Dobson circulation) in the used multiple linear regression model in Weber et al. (2018), but might be linked to the strong Arctic ozone loss in 2011 (Manney et al., 2011). The below-average annual mean Uccle and NH total ozone in 2016 is partly ascribed to the severe Arctic ozone depletion in the same year and related to the anomalous quasi biennial oscillation (QBO) induced meridional circulation

changes (see references in Weber et al., 2018).

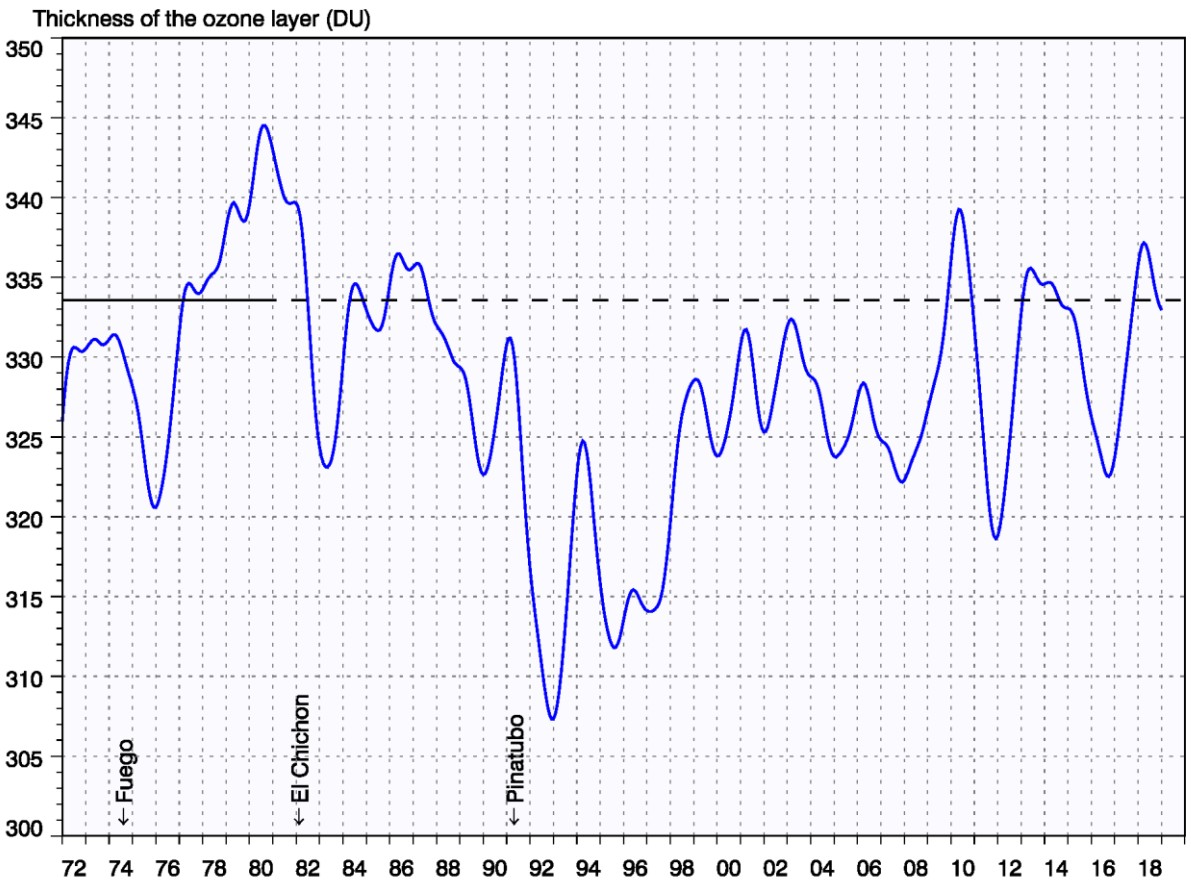

**Figure A1: Evolution of the total ozone column at Uccle as observed with Dobson D40 (1972-1989), Brewer no. 16 (1990-2001), and Brewer no. 178 (2002-present). The horizontal black full line marks the 1972-1980 total ozone average (period before catalytic ozone loss), extended until the end of the time series by the dashed horizontal line. The periods of major volcanic eruptions**
**affecting the ozone layer are indicated on the time axis as well.**

To study the long-term temporal variability of the total ozone amounts at Uccle, we make use of the LOTUS MLR regression model that we also applied to estimate the stratospheric vertical ozone trends in Sect. 4.1. The model fit and the different contributors are shown in Fig. A2. The interannual variability is reasonably captured by this model, although this MLR is not able to model the large excursions in some years, e.g. 2011-2012, without the use of some additional terms

accounting for the Arctic Oscillation or the Brewer-Dobson circulation (Weber et al., 2018). As can be noted from the observation-model residuals, the long-term temporal variability is well described by the two independent linear trends. Before 1997, ozone declined at Uccle at a rate of -1.6±0.5 % dec$^{-1}$ due to the anthropogenic production of ozone depleting substances (ODS), transported into the stratosphere, with peak concentrations in 1997. This decline rate is comparable to the NH mid-latitude value of -2 to -3% dec$^{-1}$ (Weber et al., 2018; WMO, 2018), especially considering that the Uccle total ozone

time series starts earlier than the used satellite total ozone time series in those assessments (from 1979). Subsequently, from 2000 onwards, the total ozone increased again at a rate of +1.9±0.8% dec$^{-1}$ at Uccle. This ozone recovery estimate is significantly larger than the NH mid-latitude trend of +0.2 to +0.5 % dec$^{-1}$ (Weber et al., 2018; WMO, 2018) and even larger than the expected NH trends from Equivalent Effective Stratospheric Chlorine (EESC) changes, which are about +1% dec$^{-1}$ (WMO, 2018). At Uccle, the total ozone amount seems to have nearly fully recovered yet, as could also be noted by looking

at the monthly anomaly time series in Fig. A2. Because the Dobson and Brewer spectrometers are calibrated regularly (see above), we have no doubts on the homogeneity of the time series. In general, according to Weber et al. (2018), the ozone increase after 2000 is not only due to the (slow) decrease in ODSs in the stratosphere, but also because of atmospheric dynamics, notably ozone transport via the strengthening Brewer–Dobson circulation. At Uccle, the strongest ozone increase since the beginning of this century took place in late winter – early spring (Feb-Apr), at a rate of 3-4% dec$^{-1}$, while the ozone

transport by the Brewer-Dobson circulation from its tropical source region poleward and downward into the lower stratosphere is strongest during wintertime (e.g. Butchart, 2014; Langematz, 2019).

While total ozone seems to have nearly fully recovered at Uccle, the stratospheric ozone amounts have not (see Fig. 2 and Fig. S5 for the monthly anomaly time series of the ozone concentrations in a layer 10 km above the tropopause height). The stratospheric ozone concentrations decreased between 1969-1996 with a rather consistent rate around -4% dec$^{-1}$ (between 5

to 20 km above the tropopause), hence larger than the total ozone decline rate. Since 2000, a recovery between +1 to +3% dec$^{-1}$ of the stratospheric ozone levels above Uccle is observed, although not significant and not for the upper stratospheric levels measured by ozonesondes. This value is comparable to the total ozone recovery rate at Uccle. To reconcile the stratospheric ozone trends from the ozonesondes with the total ozone trends at Uccle[1], it should also be noted that, throughout the entire free troposphere (contributing for about 10% to the total ozone amount), a very consistent increase of

the ozone concentrations at +2 to +3 % dec$^{-1}$ is measured since both 1969 and 1995 (see Fig. 3).

---

[1] Note that the ozone measurements with the ozonesondes at Uccle are normalized (dependent on the pressure and temperature) to the total ozone measurements from the co-located spectrophotometers.

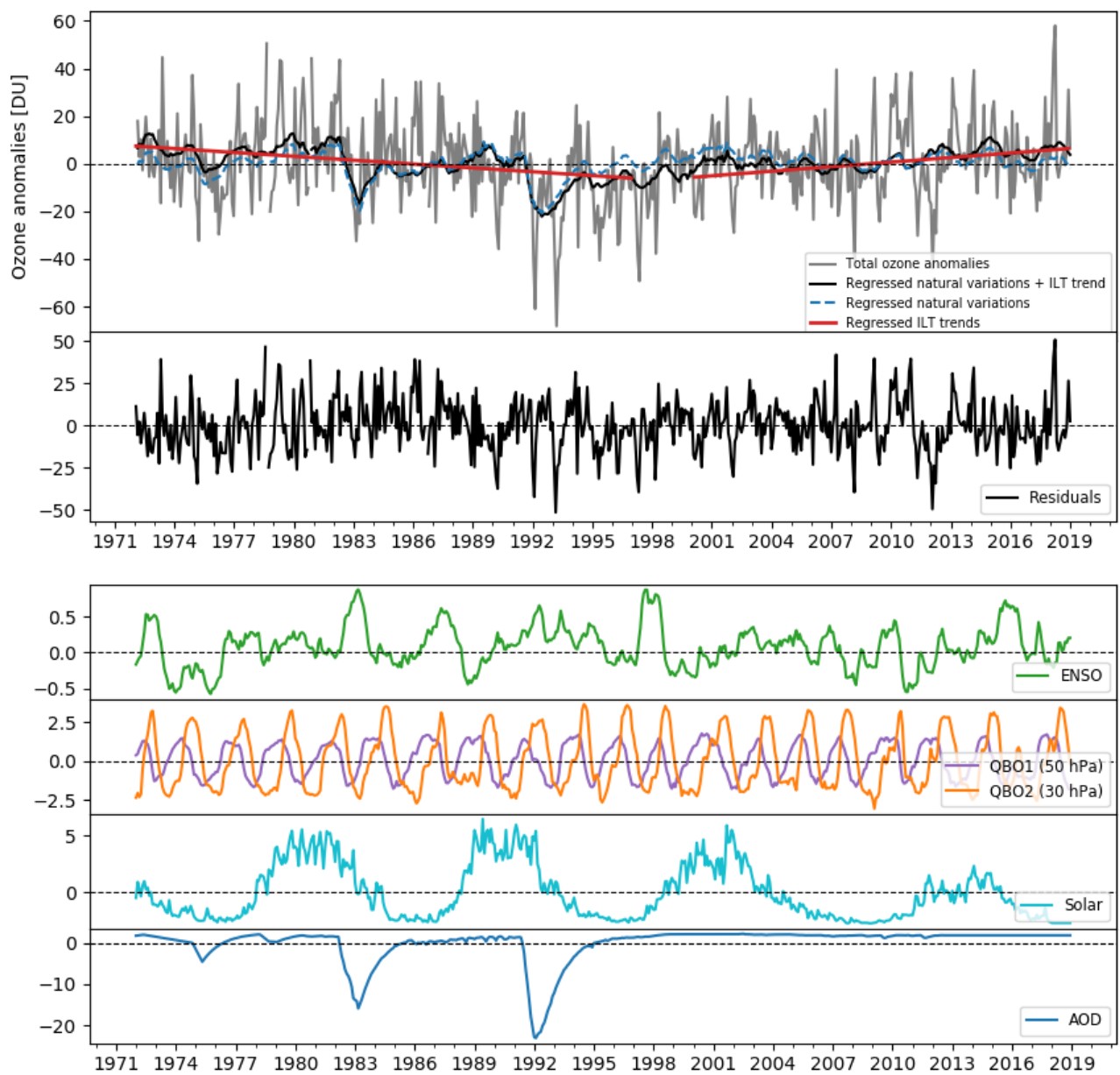

**Figure A2: Terms in the LOTUS multiple linear regression model for the Uccle total ozone amounts. The top panel shows the observed total ozone monthly anomalies in grey. The black line is the result of the full LOTUS regression model including the independent linear trends (ILTs, thick red lines). The pre-1997 trend equals -1.6 ± 0.5% dec[-1], the post-2000 trend +1.9 ± 0.8% dec[-1].The dashed blue line shows the sum of the terms of the LOTUS model without the ILTs included. The middle panel shows the residuals in the observed ozone concentrations with the full LOTUS model subtracted. The bottom panel shows the contributions of (from top to bottom) the ENSO, QBO, solar cycle, and aerosols to the reconstructed time series (blue dashed line) in the top panel.**


## Appendix B: Surface ozone trends at Uccle

In this appendix, we elaborate more on how representative and complementary the surface ozone trend derived from the ozonesonde data at Uccle is, compared to the one from a surface monitoring station at the same site. The ground network of (air quality) stations provides surface ozone measurements at higher temporal and horizontal resolution, and with higher accuracy than ozonesonde measurements, but these latter provide vertical ozone profiles in the lower troposphere as well, and sometimes even over a longer time span. As a matter of fact, the ozonesonde launch site at the urban background site Uccle also hosts surface measurements of ozone since 1986, performed by the Brussels Environment Agency. From the surface measurements, we consider the (half-hourly averaged) values at 11h30 UT, closest to the ozonesonde launch time. The monthly mean time series of those surface measurements are shown in Fig. B1, together with the lowest 1 km mean ozone measurements derived from the ozonesondes. The agreement between the surface ozone measurements from both devices is, in terms of monthly means, excellent, apart from a more or less constant offset. This offset might be explained by the difference in air masses for which the ozone concentrations are measured (surface vs. surface to 1 km above the ground), and by some Uccle pre-launch procedure of testing the ozonesonde-interface-radiosonde configuration by exposing the ozonesonde shortly (< 30s) to (stratospheric) ozone concentrations between 15 to 30 minute prior to launch. Because of the slow time constant of 20-25 minutes in the chemical reactions in the cell, this pre-launch ozone exposure might still contribute to the measured cell current immediately after launch, resulting in a positive bias in the boundary layer ozone measurements with the ozonesondes.

Both time series reveal a statistically significant (according to Spearman's test, see e.g. Lanzante, 1996) increase in surface ozone concentrations since 1986 (see Fig. B2), with a trend value 25% higher for the surface ozone measurements compared to the sonde lowest 1 km measurements (0.47 vs. 0.38 $\mu g\ m^{-3}\ yr^{-1}$ in absolute terms). Uccle is a suburban site, so, its increase in mean surface ozone concentrations is in line with the findings from Yan et al. (2018) over European suburban and urban stations during 1995–2012[2], with trends between 0.20–0.59 $\mu g\ m^{-3}\ yr^{-1}$. For the 1995-2018 time period, the ozonesonde trend (0.41 $\mu g\ m^{-3}\ yr^{-1}$, see also green curve in Fig. 3 for relative trend) is more elevated than the surface ozone trend (0.28 $\mu g\ m^{-3}\ yr^{-1}$ or 6.4±2.9 % $dec^{-1}$), and both are statistically significant. This former ozonesonde trend estimate equals the value for the entire ozonesonde time series 1969-2018 (0.39 ± 0.07 $\mu g\ m^{-3}\ yr^{-1}$), as was the case for the entire tropospheric ozone trends (see again Fig. 3).

---

[2] For comparison, over the same period, the Uccle surface ozone trend is 0.37 ± 0.20 $\mu g\ m^{-3}\ yr^{-1}$, but only 0.07 ± 0.23 $\mu g\ m^{-3}\ yr^{-1}$ for the ozonesonde measurements.

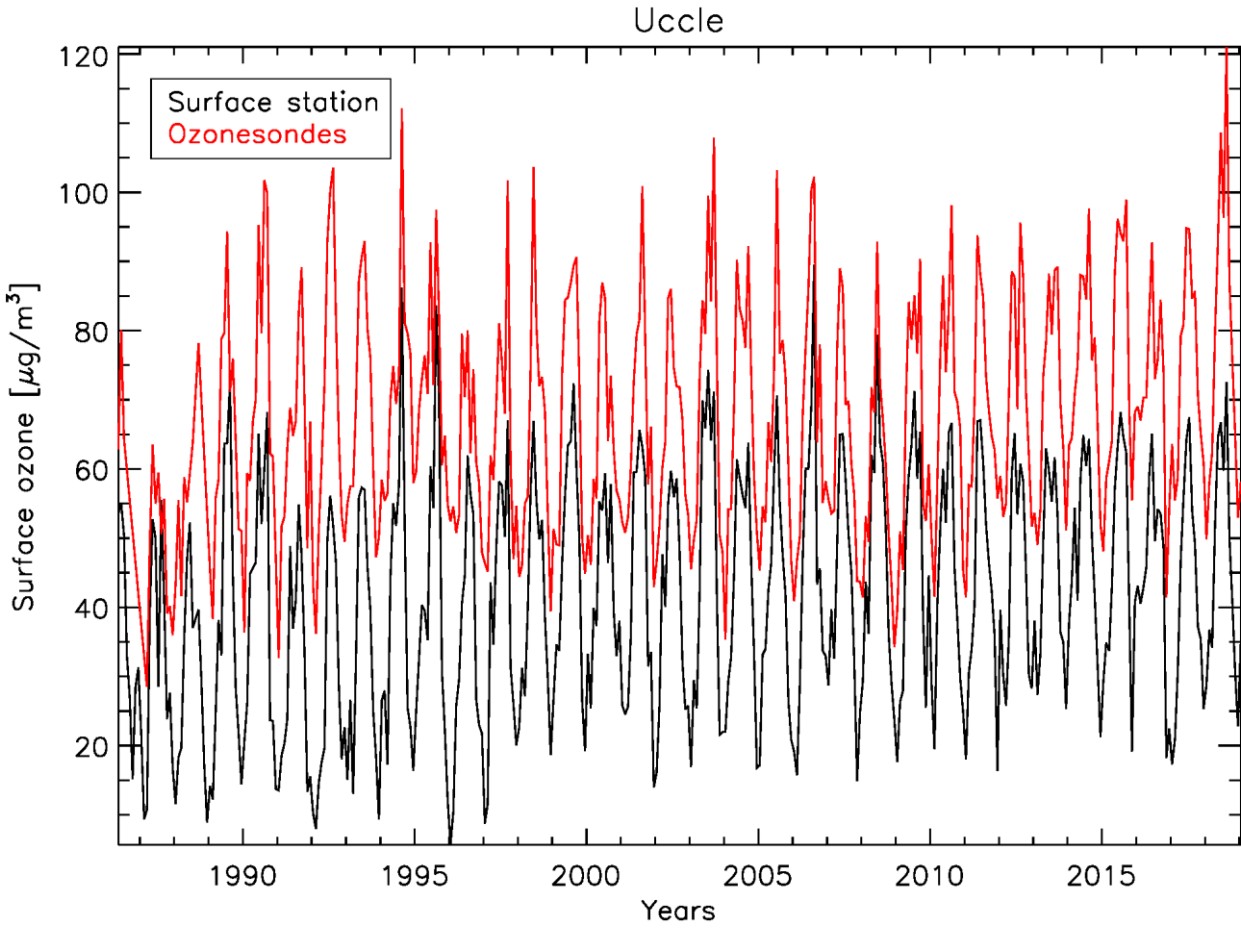

**Figure B1: Monthly mean time series of Uccle surface ozone (black) from the air quality monitoring station and mean ozone in the lowest 1 km above Uccle from the ozonesonde launches (red).**

In general, trends in surface ozone measurements are ascribed to changes in local and regional anthropogenic precursor emissions, in natural ozone precursors and/or their sources, in meteorology and weather regimes, or long-range transport patterns due to e.g. climate change (Monks et al., 2015, Lefohn et al., 2018). As ozone precursor concentrations are measured at (NO and $NO_2$) or near the Uccle site (CO measurements are available from a nearby urban traffic location at Elsene, < 5 km), we concentrate here on a possible link between the changes of those precursor mean concentrations (see

Fig. B2) to the positive surface ozone trend. Apparently, there seems to be a mismatch between the increase in ozone concentrations and the strong decreases of all available measured ozone precursor concentrations, also reported in other studies (e.g. Tørseth et al., 2012; Lefohn et al., 2018). However, it should be noted first that the photochemical production of tropospheric ozone also involves reactions implying volatile organic compounds (VOCs) and hydroxyl radical oxidation of methane and non-methane hydrocarbons, in the presence of nitrogen oxides (Monks et al., 2015). Unfortunately, those

components are not measured at the Uccle site. Moreover, the observed $NO_x$ decreases in Fig. B2 at the Uccle site can have a

reverse impact on the surface ozone trends, depending on the $NO_x$ (and VOC) concentrations. In $NO_x$ limited conditions (i.e. rural locations, but also at times of high photochemical activity on hot sunny summer days), a long-term reduction in $NO_x$ emissions lead to a surface ozone decrease. In polluted or urban areas with large $NO_x$ emissions (VOC or radical-limited conditions), or under conditions of lower photochemical activity like night-time hours, cloudy days, in wintertime,

decreasing $NO_x$ concentrations can increase ozone, also because ozone titration by NO is reduced (Lefohn et al., 2018 and references therein). Furthermore, the ozone trends also depend heavily on the chosen ozone metric (Lefohn et al., 2018). Here, we used the monthly means of the 11h30 UTC values, because the ozonesondes are launched around this time, which is a very limited frequency for surface ozone measurements. Making use of full frequency (at least hourly) of surface ozone measurements, e.g. Tørseth et al. (2012) and Lefohn et al. (2018) reported that the large $NO_x$ emission reductions that have

occurred in the past several decades in the European Union (EU) have led to a compression of the ozone distribution, where the high levels shift downward (reduced ozone peak concentrations) and the low levels shift upward. These trends are actually observed for sites in Brussels (Paoletti et al., 2014) and for the Uccle site (see Fig. S10), although there seems to be a levelling off in those opposite trends for low and high ozone concentrations since 2000 compared to the decade before (see again Fig. S10).

To conclude, explaining the increasing mean surface ozone amounts in combination with the decreasing ozone precursor emissions at Uccle is less straightforward than the (opposing) trends in high and low level ozone concentrations due to the compression of the surface ozone distribution. The interpretation of the increasing mean surface ozone concentrations is hampered by the interplay of many factors such as meteorology and transport, the non-linear dependence of the ozone concentrations on the emissions of VOC and $NO_x$, the dual role of NOx as ozone source or sink depending on the season,

and the amount of $NO_x$ emissions.

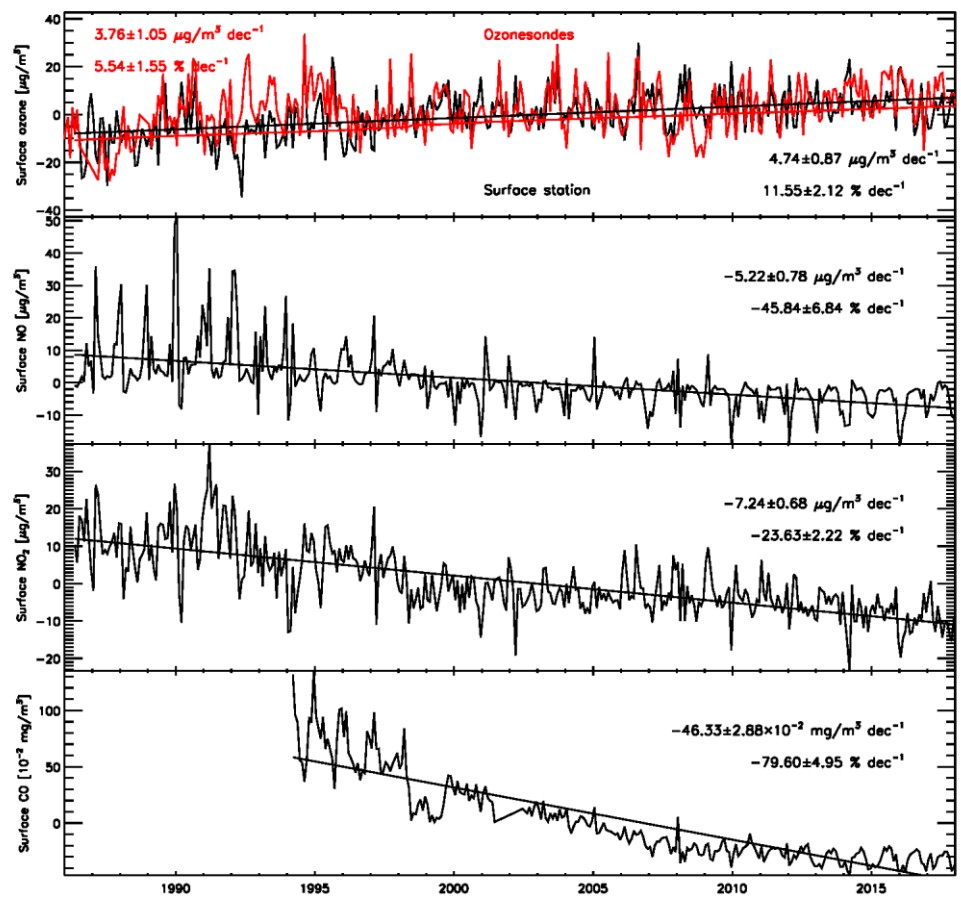

**Figure B2: Monthly anomaly time series of Uccle surface ozone (upper panel, black) and mean ozone in the lowest 1 km above Uccle from the ozonesonde launches (upper panel, red) and ozone precursor measurements at Uccle (NO, NO₂) and Elsene (CO, 5 km from Uccle). Linear trends are shown, together with the absolute and relative trend estimates, and their 2σ uncertainties.**

### Code/Data availability

The ozonesonde and total column ozone data used in this paper is publicly available through the World Ozone and Ultraviolet Radiation Data Centre (WOUDC) and the Network for the Detection of Atmospheric Composition Change (NDACC). The MOZAIC/CARIBIC/IAGOS data are available at http://www.iagos.fr and the surface ozone and ozone precursor data at Uccle can be found at http://www.irceline.be, the website of IRCEL-CELINE (Belgian Interregional Environment Agency). The AURA MLS v4.2 Uccle overpass data were obtained at http://avdc.gsfc.nasa.gov/pub, the TES data at https://search.earthdata.nasa.gov/. The source code of the LOTUS regression model is publicly available at https://arg.usask.ca/docs/LOTUS_regression.

## Author Contribution

RVM prepared the manuscript, with contributions from all authors. DDM wrote and took the lead of Sect. 2, HDB wrote Sect. 2.2.3 and made Fig. A1. DDM and HDB developed the ozonesonde data processing method and tools. DP performed the analysis for Sect. 4.1 and 5.2, and wrote Sect. 5.2. WWV wrote and did the analysis for Sect. 5.3 and helped in preparing Sect. 4.2, Sect. 5.2, and Appendix B. VDB performed part of the analysis in Sect. 4.2 and wrote that part. AD wrote and did the analysis for Sect. 5.1. MA provided the De Bilt ozonesonde dataset and gave feedback. FF provided the surface ozone and ozone precursor data at Uccle, prepared Fig. S10, and helped in preparing Sect. 4.2 and Appendix B. VT gave guidance on the use of the IAGOS data at Frankfurt airport. All authors provided comments on the manuscript.

## Competing Interests

The authors declare that they have no conflict of interest.

## Acknowledgements

The Uccle ozone sounding time series could only be built up thanks to the efforts and dedication of the ozone sounding operators over the past 50 years (Martin Lebrun, Jean-Claude Grymonpont, André Massy, Jozef Bartholomees, Daniel Wattez, Eli Weerts, Kevin Knockaert, and Roger Ameloot) and the technical support by Geert Desadelaer. Since 2007, the ozone sounding program in Uccle and R. Van Malderen are funded by the Solar-Terrestrial Centre of Excellence (STCE), a research collaboration established by the Belgian Federal Government through the action plan for reinforcement of the federal scientific institutes. We are grateful to WOUDC and the NDACC for archiving the (Uccle) ozone data and making them publicly available. The MOZAIC/CARIBIC/IAGOS data were created with support from the European Commission, national agencies in Germany (BMBF), France (MESR), and the UK (NERC), and the IAGOS member institutions (http://www.iagos.org/partners). The participating airlines (Lufthansa, Air France, Austrian, China Airlines, Iberia, Cathay Pacific, Air Namibia, and Sabena) supported IAGOS by carrying the measurement equipment free of charge since 1994. The data are available at http://www.iagos.fr thanks to additional support from AERIS. We would like to thank our colleagues from the panel for Assessment of Standard Operating Procedures for Ozonesondes (ASOPOS) for many constructive discussions about the functioning and exploitation of ozonesonde data, and in particular Herman Smit, chair of this panel. We are indebted to Daniel Zawada at the University of Saskatchewan, Canada, for his help with implementing the LOTUS regression model. We also thank Daan Hubert from the Royal Belgian Institute for Space Aeronomy for some comments and feedback on an earlier version of the manuscript.

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
