# Peer review of "years of balloon-borne ozone profile measurements at Uccle, Belgium: short history, scientific relevance and achievements in understanding the vertical ozone distribution"

_Atmospheric Chemistry and Physics, 2020_

## Referee Comment (RC1) · Anonymous Referee #3 · 4 Apr 2021

This manuscript provides a detailed overview of the ozonesonde measurements Uccle. This is one of the most important and longest records of profiles measured at a higher frequency (3/week) than nearly all other global ozonesondes sites. The manuscript covers the ozonesonde history, editing techniques applied to past data (homogenization) and includes data analysis of long-term tropospheric and stratospheric ozone. The fully homogenized data is used in the linear regression and well-documented LOUTS models to evaluate trends. Trends are evaluated and compared to MOZAIC/IAGOS commercial aircraft profiles. An additional interesting topic is the evaluation showing an increase in the frequency of tropopause fold events observed in the ozonesonde record. The manuscript also discusses and documents the changes in manufacturer sonde models used over the long record and operating procedures (changes were minimal making homogenization of long-term data much more straightforward). It also presents an important documentation of the homogenization method used for their long term records. The manuscript presents satellite comparisons/validations with the Uccle ozonesonde database, one of the most critical applications of ozonesonde data. This manuscript is a substantial contribution that shows the importance of 50 years of ozonesonde data records.

Scientific Question: Figures S6 shows and example of a tropopause fold event - a narrow high ozone layer at 600 hPa. I am not all that familiar with tropause folds but have seen some examples showing massive ozone in broad layers near the tropopause. The RH is very low in the green line (very hard to see the RH scale in light green) which would indicate stratospheric source but wondering it anything else that shows this is a purely stratospheric ozone peak?

Technical Corrections/Suggestions: Line 43: First sentence states ozone is found mainly from surface to top of atmosphere (50km) which is true for all gases. Would be good to separate it out a little more and note that Ozone, O3, is a key trace gas in the Earth's atmosphere, where is present in the troposphere but mainly resides in the lower to middle stratosphere ($\sim$90%).

Line 109: I would say something like "funding limitations or reductions" here rather than "financial problems"

Line 151: reduction by 100% may sound like the ozone signal is zero, which can be the case in very high SO2 that exceeds ozone concentration. It would be more clear to state that – in particular, SO2 reduces the ECC cell response on a 1:1 basis for every SO2 molecule.

Line 170: Change "double soundings" to "dual soundings" so it matches with text in Figure 2.

Line 208: Change "From 1990" to "Since 1990"

Line 213: SECTION 4 Note: This section title is "Temporal evolution of vertical ozone concentrations at Uccle" but the section begins with "Total ozone trends from Dobson and Brewer". Therefore, this section may be better titled as "Temporal evolution of total column and vertical ozone concentrations at Uccle"

Line 227: "...on the Uccle total ozone concentrations pops up." to " ...shows in the significant dips in Uccle total ozone."

Line 231: "(e.g. the excess total ozone in 2010, the 2011 and 2016 low ozone anomalies)." to "(e.g. the excess total ozone in 2010, and the low ozone anomalies in 2011 and 2016)."

Line 351: Improve sentence to make clear if proxies were not used: "....,as there is no consensus on the used proxies to account for natural variability" which implies that proxies were used or change to "....., as there is no consensus in using proxies to account for natural variability."

Line 368: Need more clarity here: "(since 1995, but the post-2000 trends have the same magnitude)" Is a flat trend (zero slope) meant for same magnitude. Also, wondering if post-2000 trends are shown in one of the graphs.

Line 621: Replace ..." is almost entirely compensated by the gain" with ..." has nearly fully recovered by the +2%/decade gain between 1997-2019"

Line 645: The most recent update by Stauffer shows the ozonesonde drop-off in TCO ranges from 3-7% was observed at 13 of 53 global stations (25%) 1/4 rather than 1/3.

Line 651: The drop-off was mentioned in line 645 – suggest removing ", as the total column ozone drop-off in a third of the ozonesonde stations (Stauffer et al., 655 2020)

made obvious"

---

## Referee Comment (RC2) · Anonymous Referee #1 · 12 Apr 2021

Review of Van Malderen et al. – 50 years of balloon-borne ozone profile measurements at Uccle, Belgium: short history, scientific relevance and achievements in understanding the vertical ozone distribution.

General Comments

In this manuscript, the authors provide an overview of the 50 year history of the ozonesonde program at Uccle.

The dataset is certainly extremely valuable to the field of atmospheric science, and everyone responsible should be congratulated on maintaining a high quality measurement program over such a long period of time. This is an impressive achievement!

The manuscript describes, in a medium level of detail, previous work that has been done to adjust for inhomogeneities and contamination, after which the authors present trends in total ozone at Uccle measured by ground-based instruments, a height resolved trend analysis of the Uccle ozone record in the stratosphere based on multi-linear regression, and separately a trend analysis for the troposphere, a trend analysis of tropospheric folding events and a trend analysis of surface ozone. The analyses also make use of a wide variety of other data sources to compare to the Uccle ozonesonde record. In the final section satellite overpasses are compared for three different instruments (or series of instruments) – GOME2, AURA-MLS and AURA-TES.

In my view, these components in themselves are all very worthwhile and very appropriate subjects for ACP.

Unfortunately I must state however, that I have major concerns with the design of the manuscript in its current form. Because of the very wide scope, none of the subjects are presented in sufficient detail to be of much use to the specific communities who are interested in them. In my view it would be preferable to resubmit the manuscript as three separate pieces of work, each one with a more specific focus allowing greater depth. Alternatively, the number of topics could be reduced to allow a more satisfactory level of detail to be given.

I appreciate that the authors have deliberately chosen to present something more along the lines of a summary of the many applications of the dataset.

Unfortunately, in my opinion, this decision is not optimal for the benefit of the readers of ACP.

(Please be clear I am not intending to be in any way negative about the value of the

dataset or the scientific competence of the authors, merely the structure of the submitted manuscript).

Old work is covered in too much detail, but still not in sufficient detail to enable it to be understood without reference to the original papers anyway. On the other hand, new work is "described" without enough detail provided however for the reader to make a proper assessment of its value. This is most evident in the discussion of tropospheric fold occurrence.

The major contribution of the work I see as being the assessment of long-term trends in the stratosphere and free troposphere (sections 4.2 and 4.3), at very high vertical resolution.

I would prefer these sections to be significantly expanded and sections 1, 2 and 3 shortened. I would also consider removing 4.1 and 4.4 and treating these topics in better detail in separate works. In my view section 4.1 is too superficial and section 4.4 too inconclusive in their current forms. The use of other data sets for comparison (eg DeBilt ozonesondes, Frankfurt IAGOS, surface ozone monitoring) is very interesting but does tend to clutter the analysis and obscure the conclusions.

Specific comments

Line 49: You should also quote WMO 2018 Chapter 3 (and chapter 4 if you want to talk about recovery of Antarctic ozone, seeing you have mentioned it just earlier).

Line 53 A paper about SHADOZ is not appropriate for a statement about the radiative forcing of ozone – the best reference for this would be the IPCC or the major papers they have used.

Lines 50-52 You should be careful about the radiative effect of ozone at different levels in the atmosphere.

Lines 61-64 This is confusing for the reader, because in the first sentence it sounds like "electrochemical ozonesondes" is being used broadly in a way which includes Brewer-

Mast, but then in the second sentence you distinguish ECC ozonesondes from Brewer-Mast.

Lines 67 Explain to the reader why you say "nowadays".

Lines 68-70 I am sure my Australian colleagues would also like you to mention Aspendale, which has data in WOUDC from as early as 1965 and has continued to the present, but with the launching site having moved somewhat (Aspendale – Laverton – Broadmeadows).

Line 82 I don't like "our" here – this makes it appear the work is intended for an internal audience.

Line 82 This is probably out of scope for the current work, but I would be very curious to know if the high frequency of launches at Uccle can be shown to lead to better results, compared to weekly or fortnightly or monthly launches? (You could easily enough calculate the uncertainties on the trends using only a subset of the data).

Line 91 Change "has been" to "was"

Line 114 – It sounds like the ozone profile is adjusted to agree with the total ozone measurement - You should state this clearly to the reader. (Particularly because it is the less common practice at ozonesonde stations these days).

Lines 119-135 Without more detail, it is not very convincing that this approach has worked very well.

Lines 119-135 I would also be curious to know if the results agree with more recent work, eg Tarasick 2020

Lines 139 – 163 This was all very important work, but the problem is that the reader has to take everything on trust, namely the effect of $SO_2$ on the Dobson, the effect on the ozonesonde, and the assumptions about the $SO_2$ concentration. You don't even show, for example, that when you put all this together the Dobson and the ozonesonde

are brought into agreement. Thus, I find this moderate level of detail quite frustrating. I would much prefer you be briefer, and just report the outcomes of the historic work (with full references of course).

Lines 163-185 Again, I this was very important work (and I think Uccle should be congratulated for studying the effects of the change away from Brewer-Mast compared to many other stations around the world which were not as careful) , however, the method is not actually explained here, you just show the results (and reproduce an old figure). As above, I think you should just mention this work and give the references.

Line 208 For comparison with other work, it would be helpful to give the size of the altitude correction specifically (ie not just the resulting difference in ozone).

Lines 221-223 How confident are you of the homogeneity of this time series? If you are going to show this figure (Figure 3), you need to discuss the calibration history of the Dobson (particularly in the pre-1977 era) and the transition from Dobson to Brewer.

Line 228 How can you be so confident about the attribution to volcanic eruptions? There are other large dips and several peaks as well of comparable size.

Lines 233-241 This section is too superficial. Your trend line has returned to the 1980 level by 2018 which is much faster than expected from ODS emissions. You seem to suggest the increase is therefore due to a large increase in the Brewer-Dobson circulation but then note the seasonality doesn't match. It is hard for the reader to have any confidence that the trend lines are meaningful.

Line 259 What is the justification for treating AOD as constant since 2012? Have there been no volcanic injections into the stratosphere since then that could have affected Uccle?

Line 267 I think this is really interesting work and I would have loved for you to show your full regression model and how well the different proxies can account for ozone variability at different altitudes at a specific site like Uccle.

Lines 274 I don't like the "etc" – you need to list the specific factors which have been identified.

Lines 276-277 Yes, this brings me to a question. Does it make sense to consider the relative height with respect to the tropopause once heights above 20 km say? Surely from 10 km above the tropopause, the absolute height is more relevant? This point might have been worth some consideration.

Lines 281-295 This is a very important result and deserves more attention.

Line 297 (Figure 4). This is a very important part of the manuscript. The error bars for the recent period are very hard for the reader to assimilate. There does seem to be an offset between DeBilt and Uccle, not just larger uncertainty.

Lines 304-305 What level of "the stratosphere"?

Lines 302-325 Unfortunately I find this discussion very "hand-wavy". There is a general discussion of results from the literature but I find it hard to discern what the specific conclusions are.

Lines 345-350 Frankfurt and Munich seem a long way from Uccle if considering boundary layer ozone – would you expect the trend to be correlated over this distance?

Line 342 (Figure 5) This is a very important figure but the way you've drawn it is unintentionally misleading. The reader's eye sees that the black curve and the red curve agree very well, and agree better than the blue, and it's easy to miss the fact the black curve is for a completely different time period. I would suggest only showing the three 1995-2018 trends on the one plot. If you want to include the 1969-1994 Uccle trend it could be a different panel.

Line 342 (Figure 5) Wouldn't a lot of the height range shown in this plot be above the troposphere at Uccle, particularly in winter?

Lines 399-403 This result seems extraordinary. Could the number of events really

have increased by a factor of eight in only fifty years? It seems very implausible, and if you want to include it you need to show much more convincing evidence. (Otherwise the reader will assume the most likely explanation is that there's a mistake in your algorithm).

Lines 412-465 Unfortunately, I have to say my honest recommendation would be to delete section 4.4 altogether. For one thing, it is not clear that the ozonesonde record adds any benefit compared to the continuous surface monitoring. (Noting that the ozonesonde data before 1985 is of limited value). There is a small offset which you don't seem to explain. Secondly, although you show an increasing trend, after a fairly lengthy discussion you are not able to reach any conclusions about its causes (in terms of changes in the precursors or meteorological conditions or some other factor).

Line 464 (Figure 7) Overall the figure is cluttered and difficult to read. Just from looking at the plot, while there appears to be a good agreement between the ozonesonde and the surface monitor, it is really just the existence of a consistent seasonal cycle. There seems to be a jump around the year 1998?

Line 484 Who is "us" in this context?

Line 503 You should explain why applying the averaging kernels makes a much larger improvement in the lower stratosphere compared to other altitudes.

Lines 505-509 This would be more convincing if you didn't just show the "before" and "after" plots – the reader is left wondering whether the degradation correction has been tuned to match the ozonesondes – perhaps you could show the degradation correction over time?

Line 531 In section 5.2 you don't use the averaging kernels but in section 5.1 you said how important it was to apply them – why is this?

537-538 The figure shows a growing bias at altitudes above 10 hPa – it seems to be a systematic effect and not just that the ozonesondes are "known to be less accurate".

Lines 545 (Figure 9) Overall I like this figure but the error bars seem strange. If the agreement from one year to the next is so good, it doesn't seem possible that the error bars could really be so large. The description in the caption is too brief and doesn't show what the error bars are really representing.

Line 551 Here, you should also let the reader know that TES was decommissioned in early 2018.

Line 552 You have already described the orbit of Aura in the MLS section.

Line 559 It seems curious that you apply a limit of 300 km in the troposphere but 100 km in the stratosphere – is this a reasonable thing to do?

Line 563 You should explain to the reader what is meant by the term "observation operator". Is it the same as an averaging kernel?

Line 580 In this case, seeing there are only "1-2" degrees of freedom in the troposphere, wouldn't it better to make a plot of the regression described in lines 581-590, rather than figure 10 as it currently stands? Then the reader could see the temporal stability.

---

## Author Comment (AC1) · 31 May 2021

**Response to Reviewer 3 of Van Malderen et al. – 50 years of balloon-borne ozone profile measurements at Uccle, Belgium: short history, scientific relevance and achievements in understanding the vertical ozone distribution.**

In this response, we included the reviewer comments in black. Specific comments are numbered. Our response are written in red, with the modifications in the manuscript in *red italic*.

This manuscript provides a detailed overview of the ozonesonde measurements Uccle. This is one of the most important and longest records of profiles measured at a higher frequency (3/week) than nearly all other global ozonesondes sites. The manuscript covers the ozonesonde history, editing techniques applied to past data (homogenization) and includes data analysis of long-term tropospheric and stratospheric ozone. The fully homogenized data is used in the linear regression and well-documented LOTUS models to evaluate trends. Trends are evaluated and compared to MOZAIC/IAGOS commercial aircraft profiles. An additional interesting topic is the evaluation showing an increase in the frequency of tropopause fold events observed in the ozonesonde record. The manuscript also discusses and documents the changes in manufacturer sonde models used over the long record and operating procedures (changes were minimal making homogenization of long-term data much more straightforward). It also presents an important documentation of the homogenization method used for their long term records. The manuscript presents satellite comparisons/validations with the Uccle ozonesonde database, one of the most critical applications of ozonesonde data. This manuscript is a substantial contribution that shows the importance of 50 years of ozonesonde data records.

Thank you very much for taking your time to review our manuscript and your positive feedback!

**General Comment**

1. Scientific Question: Figures S6 shows and example of a tropopause fold event - a narrow high ozone layer at 600 hPa. I am not all that familiar with tropopause folds but have seen some examples showing massive ozone in broad layers near the tropopause. The RH is very low in the green line (very hard to see the RH scale in light green) which would indicate stratospheric source but wondering if anything else that shows this is a purely stratospheric ozone peak?
   This is an example of a tropopause fold illustrating that stratospheric air can penetrate deeply in the troposphere, not only near the tropopause. The very low RH is one indicator of the stratospheric source, but the detection algorithm looks especially for tropopause folds which occur in connection to upper tropospheric frontogenesis in the polar jet stream region, as those are considered to be responsible for a large part of the mass exchange across the tropopause (see Van Haver et al., 1996, and references therein). During a test period, those authors used cross sections of potential vorticity from ECMWF analysis to check the necessary folding of the dynamical tropopause, resulting in the tuning of the stability, wind speed and vertical shear conditions as used in the detection method. We added to the figure caption that *"These criteria primarily focus on the detection of tropopause folds that occur in connection to upper tropospheric frontogenesis in the polar jet*

*stream region, as those are considered to be responsible for a large part of the mass exchange across the tropopause (see Van Haver et al., 1996, and references therein)."*

We must admit that at our printed out copies, we have no issue with the visibility of the light green color, which has also been used in other figures in the manuscript (e.g. Figs. 3, S1, S6). But, we will keep an eye on it.

**Technical Corrections/Suggestions:**

2. Line 43: First sentence states ozone is found mainly from surface to top of atmosphere (50km) which is true for all gases. Would be good to separate it out a little more and note that Ozone, O3, is a key trace gas in the Earth's atmosphere, where is present in the troposphere but mainly resides in the lower to middle stratosphere (90%).
   Done, we added: *"with the highest concentrations in the lower to middle stratosphere (90% of total column ozone amount)."*

3. Line 109: I would say something like "funding limitations or reductions" here rather than"financial problems"
   Done, changed to *"funding reductions".*

4. Line 151: reduction by 100% may sound like the ozone signal is zero, which can be the case in very high SO2 that exceeds ozone concentration. It would be more clear to state that – in particular, SO2 reduces the ECC cell response on a 1:1 basis for every SO2 molecule.
   You are right, we changed it to *"In particular, one $SO_2$ molecule cause a reverse current of two electrons, reducing the electrochemical cell response on a 1:1 basis"*

5. Line 170: Change "double soundings" to "dual soundings" so it matches with text in Figure 2.
   Done as suggested.

6. Line 208: Change "From 1990" to "Since 1990"
   Done as suggested.

7. Line 213: SECTION 4 Note: This section title is "Temporal evolution of vertical ozone concentrations at Uccle" but the section begins with "Total ozone trends from Dobson and Brewer". Therefore, this section may be better titled as "Temporal evolution of total column and vertical ozone concentrations at Uccle"
   As a response to the other review report, we moved the "Total ozone trends" subsection to an appendix, so the title of Section 4 now better covers the entire contents of this section.

8. Line 227: "...on the Uccle total ozone concentrations pops up." to "...shows in the significant dips in Uccle total ozone."
   Done, changed to *"is shown in the significant dips in Uccle total ozone."*

9. Line 231: "(e.g. the excess total ozone in 2010, the 2011 and 2016 low ozone anomalies)." to "(e.g. the excess total ozone in 2010, and the low ozone anomalies in 2011and 2016)."
   Done as suggested.

10. Line 351: Improve sentence to make clear if proxies were not used: "....,as there is no consensus on the used proxies to account for natural variability" which implies that

proxies were used or change to "....., as there is no consensus in using proxies to account for natural variability."

*Done, changed to "as there is no consensus in using (which) proxies to account for natural variability."*

11. Line 368: Need more clarity here: "(since 1995, but the post-2000 trends have the same magnitude)" Is a flat trend (zero slope) meant for same magnitude. Also, wondering if post-2000 trends are shown in one of the graphs.
    You are right. We changed the text in "*The Uccle tropospheric ozone concentrations have been increasing at about the same rate since 1969 (in black in Fig. 3) as since 1995 (in green in Fig. 3), and also the post-2000 increase rate is very similar (not shown here, but could to some extent be noted from the tropospheric ozone column time series shown in Fig. S7).*"

12. Line 621: Replace..." is almost entirely compensated by the gain" with..." has nearly fully recovered by the +2%/decade gain between 1997-2019"
    Done as suggested.

13. Line 645: The most recent update by Stauffer shows the ozonesonde drop-off in TCO ranges from 3-7% was observed at 13 of 53 global stations (25%) 1/4 rather than 1/3.
    Yes, thank you, we are aware of this update (being member of the ECC Drop-off Task Team within ASOPOS) and even a more recent one (after re-processing of two Canadian sites). We added to the text: *"a number now reduced to about 20% (12 of 60 global stations, Stauffer et al., 2021, private communication)".*

14. Line 651: The drop-off was mentioned in line 645 – suggest removing ", as the total column ozone drop-off in a third of the ozonesonde stations (Stauffer et al., 2020) paper made obvious"
    Done as suggested.

---

## Author Comment (AC2) · 31 May 2021

**Response to Reviewer 1 of Van Malderen et al. – 50 years of balloon-borne ozone profile measurements at Uccle, Belgium: short history, scientific relevance and achievements in understanding the vertical ozone distribution.**

In this response, we included the reviewer comments in black. Specific comments are numbered. Our response are written in red, with the modifications in the manuscript in *red italic*.

**General Comments**

In this manuscript, the authors provide an overview of the 50 year history of the ozonesonde program at Uccle.

The dataset is certainly extremely valuable to the field of atmospheric science, and everyone responsible should be congratulated on maintaining a high quality measurement program over such a long period of time. This is an impressive achievement!

The manuscript describes, in a medium level of detail, previous work that has been done to adjust for inhomogeneities and contamination, after which the authors present trends in total ozone at Uccle measured by ground-based instruments, a height resolved trend analysis of the Uccle ozone record in the stratosphere based on multi-linear regression, and separately a trend analysis for the troposphere, a trend analysis of tropospheric folding events and a trend analysis of surface ozone. The analyses also make use of a wide variety of other data sources to compare to the Uccle ozonesonde record. In the final section satellite overpasses are compared for three different instruments (or series of instruments) – GOME2, AURA-MLS and AURA-TES.

In my view, these components in themselves are all very worthwhile and very appropriate subjects for ACP.

Unfortunately I must state however, that I have major concerns with the design of the manuscript in its current form. Because of the very wide scope, none of the subjects are presented in sufficient detail to be of much use to the specific communities who are interested in them. In my view it would be preferable to resubmit the manuscript as three separate pieces of work, each one with a more specific focus allowing greater depth. Alternatively, the number of topics could be reduced to allow a more satisfactory level of detail to be given.

I appreciate that the authors have deliberately chosen to present something more along the lines of a summary of the many applications of the dataset.

Unfortunately, in my opinion, this decision is not optimal for the benefit of the readers of ACP.

(Please be clear I am not intending to be in any way negative about the value of the dataset or the scientific competence of the authors, merely the structure of the submitted manuscript).

Old work is covered in too much detail, but still not in sufficient detail to enable it to be understood without reference to the original papers anyway. On the other hand, new work is "described" without enough detail provided however for the reader to make a proper assessment of its value. This is most evident in the discussion of tropospheric fold occurrence.

The major contribution of the work I see as being the assessment of long-term trends in the stratosphere and free troposphere (sections 4.2 and 4.3), at very high vertical resolution.

I would prefer these sections to be significantly expanded and sections 1, 2 and 3 shortened. I would also consider removing 4.1 and 4.4 and treating these topics in better detail in separate works. In my view section 4.1 is too superficial and section 4.4 too inconclusive in their current forms. The use of other data sets for comparison (eg De Bilt ozonesondes, Frankfurt IAGOS, surface ozone monitoring) is very interesting but does tend to clutter the analysis and obscure the conclusions.

Thank you very much for your honest review report, and for pointing your finger to some very important issues on specific analyses.

Concerning your remarks about the design of the manuscript. We are aware that, already in your quick review report, you raised the same concern of having too much specific topics, which had **partially** been picked up by the handling associate editor. Therefore, we contacted first the editor for additional advice before starting a complete reorganization of the manuscript. We repeat here again our arguments and inspiration behind the manuscript.

First of all, the two following ACP papers inspired us for our approach (presentation and organisation of the manuscript):

- Staehelin, J., Viatte, P., Stübi, R., Tummon, F., and Peter, T.: Stratospheric ozone measurements at Arosa (Switzerland): history and scientific relevance, Atmos. Chem. Phys., 18, 6567–6584, https://doi.org/10.5194/acp-18-6567-2018, 2018.
- De Mazière, M., Thompson, A. M., Kurylo, M. J., Wild, J. D., Bernhard, G., Blumenstock, T., Braathen, G. O., Hannigan, J. W., Lambert, J.-C., Leblanc, T., McGee, T. J., Nedoluha, G., Petropavlovskikh, I., Seckmeyer, G., Simon, P. C., Steinbrecht, W., and Strahan, S. E.: The Network for the Detection of Atmospheric Composition Change (NDACC): history, status and perspectives, Atmos. Chem. Phys., 18, 4935–4964, https://doi.org/10.5194/acp-18-4935-2018, 2018.

In this sense, we believe that this manuscript has only one focus or topic, i.e. the Uccle ozonesonde measurements, with the aim to demonstrate the scientific relevance of and the major achievements with this dataset. As ozonesondes are still the only technique able to measure the ozone concentrations all the way up from the surface to the middle stratosphere with very high accuracy and vertical resolution, they have many application areas in which they are crucial: (i) long-term variability in stratospheric and tropospheric ozone, (ii) backbone for satellite validation, with the satellites mostly measuring ozone only in stratosphere or upper troposphere, (iii) process studies in stratospheric-tropospheric exchange, and chemical production/destruction of ozone. We believe that the strength and uniqueness of the ozonesonde measurements, and in particular of our long-term and very dense Uccle dataset, lies precisely in combining all those different aspects of ozone research. To our opinion, focusing on only one aspect (e.g. trends) really underrate the

relevance of our dataset. And therefore we believe that it is better to see this manuscript as a whole, rather than break it in pieces.

These were the 2 options we proposed to the editor for a new organization of the paper:

1) The current one, which was already a reorganization and more focused one as the original manuscript (to which your quick report was addressed).
2) In case the proposed option one still contains too much topics according to you, we propose to split up the paper in two (and not three) parts, namely "50 years of balloon-borne ozone profile measurements at Uccle, Belgium. Part 1: Major achievements for stratospheric ozone" & "50 years of balloon-borne ozone profile measurements at Uccle, Belgium. Part 2: Major achievements for tropospheric ozone". The exact titles of the different parts still need to be better determined, but the idea behind is to keep the focus on the measurement dataset by publishing the analysis as a series of two papers. The stratospheric part would then contain most of the short history, the total and stratospheric ozone trends and the AURA-MLS (stratospheric ozone satellite retrieval) comparison. The tropospheric part would contain some aspects of the short history (e.g. correction for SO2), the tropospheric and surface ozone trends (with increased frequency of tropopause folds), and the AURA-TES (tropospheric ozone satellite retrieval) comparison. In this sense, both the stratospheric and tropospheric ozone community are served separately, the two papers will be shorter and lighter than the combined one and we keep only two topics in each of the more focused papers. If this option is the preferred one, we will submit the two papers at the same time, because they belong together. Of course, there will be inevitably cross-references between both manuscripts.

The editor had a preference for the current organization (option 1) of the manuscript, which was then accepted in ACPD.

From your comments, we understand that you believe that the assessment of long-term trends in the stratosphere and free troposphere (sections 4.2 and 4.3), at very high vertical resolution, is the major contribution of the work. However, you mention that this contribution is cluttered by taking the total and surface ozone trends at Uccle and the De Bilt ozonesonde and IAGOS Frankfurt measurements into account. We think that the opposite is true, and that these measurements underline the broader perspective, relevance and (spatial) representativeness of the Uccle ozonesonde measurements. A study presenting the vertical ozone trends at one single location only is less relevant than when these trends are compared with nearby, independent measurements. Nevertheless, we moved the Dobson/Brewer total ozone trends, making use of the LOTUS multiple linear regression model, at Uccle to Appendix A, and the surface ozone trends at Uccle to Appendix B.

We also do not fully agree with you that sections 1, 2, 3 contain too much detail and section 4 (the trends) not enough. For instance, in the section on the history, we only shortly describe the three challenges the Uccle time series has been faced with (among others) that made the fame of the Uccle time series (frequency response analysis, $SO_2$ correction, BM to ECC transition) and are still very relevant for current new developments in ozonesonde data processing (frequency response analysis) or essential when calculating trends with the ozonesonde record over the entire time series (BM to ECC transition) or over subperiods ($SO_2$ correction). As all those analyses have already been published elsewhere, we give all

the needed references for a reader that wants to have all the details. We only shortly describe the research and the main outcome of these past studies for this paper. We also do not fully understand the criticism that new work (the trends in section 4) is only described without enough detail and that the analysis is inconclusive or too superficial, especially taking into consideration that you proposed not including the De Bilt and IAGOS Frankfurt measurements. Every trend result is put in a broader perspective and compared to the most recent findings in other studies and trend assessments (often based on satellite ozone retrievals), while the interpretation is fed by references to recent model results. To give some examples: for the explanation of all specific features in the total ozone time series, we refer to Fig. 2 in Weber et al. (2018), we compare the lower-stratospheric ozone trends with the Ball et al. (2018, 2019, 2020) studies, we refer to all relevant surface and tropospheric ozone studies above Europe and confront their findings with ours. Indeed, some findings are inconclusive (why is the total ozone recovery at Uccle almost fully accomplished in contrast to global total ozone recovery? Why do we see the opposite lower-stratospheric ozone trends as in the Ball et al. papers? What processes are responsible for the surface/tropospheric ozone trends at Uccle?), but this is related to the fact that we only treat with great detail one station (first two questions raised), and that findings are still inconclusive on the global scale (third question raised). However, in answer to your specific comments, we tried to make the analysis more detailed, conclusive, and in-depth, where possible.

**Specific comments**

1. Line 49: You should also quote WMO 2018 Chapter 3 (and chapter 4 if you want to talk about recovery of Antarctic ozone, seeing you have mentioned it just earlier).
   Done as suggested.

2. Line 53 A paper about SHADOZ is not appropriate for a statement about the radiative forcing of ozone – the best reference for this would be the IPCC or the major papers they have used.
   We changed it to "*estimated to have contributed ~20% as much positive radiative forcing as $CO_2$ since 1750 (IPCC, 2013)*", with text directly taken from the IPCC report.

3. Lines 50-52 You should be careful about the radiative effect of ozone at different levels in the atmosphere.
   We specified that ozone can act as a greenhouse gas at certain altitudes.

4. Lines 61-64 This is confusing for the reader, because in the first sentence it sounds like"electrochemical ozonesondes" is being used broadly in a way which includes Brewer-Mast, but then in the second sentence you distinguish ECC ozonesondes from Brewer-Mast.
   We believe the text is correct. Both BM and ECC sondes are electrochemical sensors: BM uses electrodes of different metal, and ECC (electrochemical concentration cell) sondes use different concentrations. We specified this further in the text.

5. Lines 67 Explain to the reader why you say "nowadays".
   We added "*Before the digital sounding systems era the vertical resolution was less due to the manual sampling technique by the operator, providing only measurements at significant levels.*"

6. Lines 68-70 I am sure my Australian colleagues would also like you to mention Aspendale, which has data in WOUDC from as early as 1965 and has continued to the present, but with the launching site having moved somewhat (Aspendale – Laverton –Broadmeadows).

Thanks for pointing this out. We added this information, and we also included the Japanese sites Tateno (Tsukuba) and Sapporo, that initiated ozone soundings in 1968 and 1969, respectively. We changed the text into "*Regular measurements with ozonesondes started in the second half of the 1960s at a few sites: in 1965 at Aspendale (Australia, but moved to other suburbs of Melbourne thereafter, i.e. Laverton and Broadmeadows), in 1966 at Resolute Bay (Canada), in 1967 at Hohenpeissenberg (Germany), in 1968 at Payerne (Switzerland) and at Tateno (Tsukuba, Japan), in 1969 at Uccle (Belgium) and Sapporo (Japan), and in 1970 at Wallops Island (USA).*"

7. Line 82 I don't like "our" here – this makes it appear the work is intended for an internal audience.

We agree, we changed "*our*" to "*the*".

8. Line 82 This is probably out of scope for the current work, but I would be very curious to know if the high frequency of launches at Uccle can be shown to lead to better results, compared to weekly or fortnightly or monthly launches? (You could easily enough calculate the uncertainties on the trends using only a subset of the data).

Note that in the manuscript, we already made a reference to the recent work by Chang et al. (2020), who, based on a sensitivity analysis with IAGOS profiles above Europe, determined that an optimal sample frequency of 14 profiles per month is required to calculate tropospheric ozone trends with their integrated fit method (and about 18 profiles a month when this method is not used).

In the figure 1 here below, we calculated the vertical ozone trends (both of the troposphere and stratosphere) for the time periods 1969-2018 and 2000-2018 with the normal Uccle sounding frequency of 3 times/week and with reduced frequency (once a week). We used here a simple linear regression model for the fit, as both stratospheric and tropospheric ozone trends have been estimated. From this figure, it should be clear that the differences between both the trends estimates and their uncertainties are small when derived from the entire and reduced frequency datasets, if the 50 year time period is considered. However, for the ozone recovery time period (after 2000), large trend differences occur, especially in the important lower-stratospheric region.

[Figure]

**Figure 1: Vertical distribution of trends of ozone concentrations at Uccle for different periods and for different sounding frequencies (3 times a week and once a week). The trends and their 2-sigma error uncertainties are calculated a simple linear regression model.**

9. Line 91 Change "has been" to "was"
   Done as suggested.

10. Line 114 – It sounds like the ozone profile is adjusted to agree with the total ozone measurement - You should state this clearly to the reader. (Particularly because it is the less common practice at ozonesonde stations these days).
    This was (and is in case of Hohenpeissenberg) common practice for BM sondes, and is essential for these ozonesondes, as they are only sensitive to about 80-90% of the measured total ozone amount during a flight. This is explained in Sect. 2.2.3, but we specified this here as well by adding *"(essential for BM ozonesondes, see Sect. 2.2.3)"*. In our operational correction/processing of the ozonesonde data, called PRESTO, we indeed apply a "pressure and temperature dependent total ozone normalization" for both the BM and ECC ozonesondes. A total ozone normalization is not needed for ECC ozonesondes (around 100% sensitive to the measured total ozone column amount during a flight), and it is advised not applying it (in e.g. the Ozonesonde Data Quality Assessment or O3S-DQA homogenization guidelines). However, when merging the BM and ECC time series for e.g. estimating trends, we rely on the PRESTO corrections, that have proven to ensure the homogeneity of the entire time series (see Sect. 2.2.3). A comparison between the PRESTO and O3S-DQA corrections for the Uccle ECC time series is available in Van Malderen et al. (2016). All this information is provided further in the manuscript.

11. Lines 119-135 Without more detail, it is not very convincing that this approach has worked very well.

We included some specifications (see next point). In the paper about the frequency response of BM sondes, the whole procedure about the measuring procedure for the determination of the frequency response by means of a Fourier transform as well as the deconvolution procedure are explained in detail. The example of the ascent and descent data of an ozone sounding before and after deconvolution shows the effectiveness of the method. Therefore it is hard to see what additional detail could make it more convincing that this approach works very well, especially with the more recent publications you are referring to in your next point, in mind. The De Muer and Malcorps (1984) paper was really the pioneer for the current ideas about convolution and deconvolution of the ozonesonde measurements (Vömel et al., 2020, and Tarasick et al., 2021).

12. Lines 119-135 I would also be curious to know if the results agree with more recent work, eg Tarasick 2020

We rewrote large parts of this section to make a more direct reference to this recent work: "*They found three different time constants: (i) a first-order process with a time constant of about 17 to 25 s (depending on the solution temperature) caused by the formation of iodine in the solution, (ii) a time constant of 7s, likely to be caused by the diffusion of iodine molecules to the platinum cathode, and (iii) a time constant of about 2.8 min that was explained by another diffusion process, i.e. an adsorption and subsequent desorption process of ozone at the surface of the air-sampling system. The slow first-order process with a time constant of about 20-25 minutes (found by Salzman and Gilbert (1959) and taken up by Vömel et al., 2020, and Tarasick et al., 2021) could not be identified, probably because the impact of this process for a 0.1% KI solution would be too small (being 10% of the fast process for a 1% KI solution), as noted in De Muer and Malcorps (1984).*"

13. Lines 139 – 163 This was all very important work, but the problem is that the reader has to take everything on trust, namely the effect of SO2 on the Dobson, the effect on the ozonesonde, and the assumptions about the SO2 concentration. You don't even show, for example, that when you put all this together the Dobson and the ozonesonde are brought into agreement. Thus, I find this moderate level of detail quite frustrating. I would much prefer you be briefer, and just report the outcomes of the historic work (with full references of course).

Yes, you are right that we rely on the reader consulting the references (we provide full references here) if he/she wants to find out more details on the effect of $SO_2$ and how the correction works. But, as we focus here on the ozonesonde measurements, we do show the impact of the $SO_2$ on an ozonesonde profile in Fig.1 and Fig. S1, and on the associated vertical ozone trends in Fig. S3 (as well as the impact of the $SO_2$ correction). Also the time variability of the $SO_2$ amounts at Uccle is shown in Fig. S2. We restructured the text in this section, to better put the emphasis of the $SO_2$ impact on ozonesonde measurements, and presented the $SO_2$ correction for Dobson total ozone measurements as a "supporting tool" for the correct evaluation of vertical tropospheric ozone trends for the 1969-1996 period, for which the combination of Brewer-Mast ozonesondes and a Dobson spectrophotometer for the necessary total ozone normalization of the BM ozonesondes, has been used. We also now mention the main idea/principle of the $SO_2$ corrections.
As to the question about the agreement between Dobson and ozonesonde data:
after any correction applied to each ozone sounding (such as the $SO_2$ correction), for Brewer-Mast sondes it was common practice to apply a final normalization to the integrated ozone amount (Dobson). This is a standard procedure that makes the $SO_2$ correction not relevant for the agreement between the two corrected data sets.

14. Lines 163-185 Again, I this was very important work (and I think Uccle should be congratulated for studying the effects of the change away from Brewer-Mast

compared to many other stations around the world which were not as careful) , however, the method is not actually explained here, you just show the results (and reproduce an old figure). As above, I think you should just mention this work and give the references.

Yes, you are right that not all the details of the method are actually explained here, only its main principles. The details of the method are already available in other papers (De Backer et al., 1998, De Backer 1999, and more recently also in Van Malderen et al., 2016), so there is no need to describe the method in full detail again. We moved the figure to the supplementary material (Fig. S4 now). But, as it is actually this (operational) method that has been used to process the 50 years of ozonesonde measurements analysed in this manuscript (see Section 3), it is essential to describe the reason behind and the rationale of the method. We made some small specifications/clarifications in the text and added the De Backer 1999 reference.

15. Line 208 For comparison with other work, it would be helpful to give the size of the altitude correction specifically (ie not just the resulting difference in ozone).

We added "*sonde altitudes were too low up to 1000m at an altitude of 30km*".

16. Lines 221-223 How confident are you of the homogeneity of this time series? If you are going to show this figure (Figure 3), you need to discuss the calibration history of the Dobson (particularly in the pre-1977 era) and the transition from Dobson to Brewer.

For your comments 16 to 18, please note that we decided to move this section to an appendix. We however still find it very relevant to include this section, as the Uccle ozonesonde time series are normalized (with a pressure and temperature dependent procedure) to the total ozone measurements with the Dobson and Brewer. Describing and explaining the vertical stratospheric ozone trends without mentioning the total ozone trends does not seem to be scientific "fair" to our opinion.

For this specific comment, we added the following sentence (and references): "*The calibration history of the Dobson instrument is documented in De Muer and De Backer (1992) and the transition to the Brewer instrument is described in De Backer and De Muer (1991). Both Brewer instruments were recalibrated against the traveling standard Brewer instrument no. 17 in 1994 (no. 16 only), 2003, 2006, 2008, and against the travelling reference Brewer no. 158 since 2010 every second year. The stability of the instruments is also continuously checked against the co-located instruments (with the Dobson no. 40 from 1991 until May 2009, between both Brewers since 2001). Internal lamp tests are performed on a regular basis to check whether a Brewer instrument is drifting. When instrumental drift is detected, it is corrected for.*"

17. Line 228 How can you be so confident about the attribution to volcanic eruptions? There are other large dips and several peaks as well of comparable size.

We changed the text to "*Indeed, the episodes of enhanced stratospheric aerosol-related ozone loss after those major volcanic eruptions are confirmed by model results (see e.g. Tie and Brasseur, 1995, Solomon, 1999,  Aquila et al., 2013 for a description of the mechanism behind) and can clearly be identified in the time series.*"
We also made some small changes when referring to Weber et al., 2018 in the following lines, so that it should be clear that those authors make the same attribution of those dips to volcanic eruptions in their analysis, based on the same arguments.

We added a description of the attribution of other large dips and peaks at the end of the time series, based on the relevant literature: "*In 2010, the Uccle ozone levels were unusually high, as over the entire NH extratropics. An unusually pronounced and persistent negative phase of the Arctic Oscillation and North Atlantic Oscillation*

*in 2010, with the co-incidence of northern winter 2009/2010 with the easterly wind-shear phase of the QBO have been identified as major contributors (Steinbrecht et al., 2011) of this excess ozone. The 2011 ozone low anomaly cannot be fully explained by including this Arctic Oscillation and other dynamical proxies (e.g. for the Brewer-Dobson circulation) in the used multiple linear regression model in Weber et al. (2018), but might be linked to the strong Arctic ozone loss in 2011 (Manney et al., 2011). The below-average annual mean Uccle and NH total ozone in 2016 is partly ascribed to the severe Arctic ozone depletion in the same year and related to the anomalous quasi biennial oscillation (QBO) induced meridional circulation changes (see references in Weber et al., 2018).*"

18. Lines 233-241 This section is too superficial. Your trend line has returned to the 1980 level by 2018 which is much faster than expected from ODS emissions. You seem to suggest the increase is therefore due to a large increase in the Brewer-Dobson circulation but then note the seasonality doesn't match. It is hard for the reader to have any confidence that the trend lines are meaningful.

Thank you for this feedback. We understand your concern. Therefore, we also applied the LOTUS MLR model to the total ozone time series at Uccle and included the fit of the LOTUS MLR regression model and its different contributing terms (or proxies) to the total ozone monthly anomalies in the appendix A (Fig. A2). However, the trend estimates from this LOTUS model also give a nearly full total ozone recovery at Uccle, which can also be seen from the monthly anomaly time series itself in Fig. A2. As the spectrophotometers at Uccle are regularly calibrated on-site, we have no doubts on the homogeneity of the time series. Additionally, those trends are not completely incompatible with the vertical ozone trends estimated from the ozonesondes (which are of course normalized, pressure and temperature dependent to the total ozone measurements). We added an extra discussion in the manuscript.

19. Line 259 What is the justification for treating AOD as constant since 2012? Have there been no volcanic injections into the stratosphere since then that could have affected Uccle?

This is the approach followed in the LOTUS regression model for the AOD, so we consistently followed it. This approach is of course only valid assuming that during those last extrapolated years, the mean AOD was representative of background values. We think that for a NH site like Uccle, this is a valid assumption, whereas for SH sites, e.g. the impact of the 2015 Calbuco eruption in Chile should not be underestimated (as it had an impact on the Antarctic ozone hole recovery e.g. Solomon et al., 2016, 10.1126/science.aae0061, and Ivy et al., 2017 https://doi.org/10.1002/2016GL071925).

20. Line 267 I think this is really interesting work and I would have loved for you to show your full regression model and how well the different proxies can account for ozone variability at different altitudes at a specific site like Uccle.

Thank you for your appreciation of this discussion. The output of the LOTUS MLR regression model and the different contributing terms (or proxies) for the monthly anomaly ozone concentrations at the layer 10 km above the tropopause (close to the ozone peak) are shown in Fig. S5. However, we think that showing the results of another multiple linear regression model for Uccle, with other proxies, does not bring a lot of added value here. First, we think that the use of a community developed and accepted MLR model should be preferred above an in-house developed MLR model. Secondly, the LOTUS report itself contains a nice overview of the sensitivity of the MLR models to the proxies for determining long-term trends in stratospheric ozone (see Section 4.3 "Sensitivity tests"). Thirdly, if we apply a simple linear regression model to calculate trends for Uccle, the estimated values are not significantly different from those of the LOTUS MLR model, as can be seen in Fig. 2 here below.

However we added "However, here, the analysis is limited to the LOTUS model *and the sensitivity of the estimated trends on the chosen (M)LR model is very low for the Uccle time series.*"

[Figure]

**Figure 2: Vertical distribution of trends of ozone concentrations at Uccle for three different periods. The trends and their 2-sigma error uncertainties are calculated a simple linear regression model.**

21. Lines 274 I don't like the "etc" – you need to list the specific factors which have been identified.
    OK, we dropped the etc. All relevant factors are already summed up.

22. Lines 276-277 Yes, this brings me to a question. Does it make sense to consider the relative height with respect to the tropopause once heights above 20 km say? Surely from 10 km above the tropopause, the absolute height is more relevant? This point might have been worth some consideration.
    In our opinion, it makes sense to still consider the relative height w.r.t. the tropopause at the higher pressure levels of an ozone sounding measurement, as we then also cancel out the seasonal variability of the ozone peak to some extent. In Fig. 3 here below, it can be seen that the ozone peak at Uccle lies at higher altitudes in summer (when the tropopause is also higher) than in wintertime (with a lower tropopause). For calculating long-term ozone trends at different pressure levels, we want to remove the seasonal cycle as much as possible, which leads to trend estimates which are more profile independent (i.e. more flat in the vertical component). To illustrate this, you can compare the vertical ozone trends from Fig. 2 and Fig. 4 in this

response here. You can observe that the shape of the ozone profile is reflected in the vertical ozone trends with absolute altitude as vertical coordinate, while this is less the case for the trends as a function of the altitude relative to the tropopause.

We added the following sentence to the text: "*However, also for these altitudes, we prefer to calculate the vertical ozone trends in altitudes relative to the tropopause, to cancel out the seasonal variation of the ozone peak altitude, which roughly follows the tropopause height variation at Uccle: the ozone maximum peak is at its highest altitudes in summer (when the tropopause is also located higher), and lies at lower altitudes in winter (with the lowest tropopause). This approach gives in general vertical ozone trends that vary less over the different altitude levels.*"

[Figure]

**Figure 3: Monthly mean ozone profiles at Uccle.**

[Figure]

**Figure 4: Vertical distribution of trends of ozone concentrations at Uccle for three different periods, but now with absolute altitude as vertical coordinate. The trends and their 2-sigma error uncertainties are calculated with a simple linear regression model.**

23. Lines 281-295 This is a very important result and deserves more attention.

    After the submission of this manuscript, a couple of studies describing and explaining the lower stratospheric ozone decline in global observations have appeared. We updated and re-organized the discussion to those latest findings. However, it should also be noted that the lower-stratospheric ozone trends calculated from two nearby ozonesonde sites should also be put in perspective to the opposing trends from global satellite ozone retrievals and chemical climate model ensembles. Only a global assessment of the lower stratospheric ozone trends from different ground-based ozone instruments, could intervene with more weight in this discussion.

24. Line 297 (Figure 4). This is a very important part of the manuscript. The error bars for the recent period are very hard for the reader to assimilate. There does seem to be an offset between De Bilt and Uccle, not just larger uncertainty.

    We increased the resolution of the figure. We have been considering other representations of the error bars (by symbols like triangles, circles) and other line styles for the lines connecting the error bars, but the figure gets very busy with this non-standard representation. We also want to keep this plot uniform with the vertical tropospheric ozone trends plot (now Fig. 3).

    Yes, you are absolutely right that there is an offset. But taking into account the uncertainties of both, you cannot state that this offset is statistically significant. Possible explanations between the vertical ozone distribution and trends between Uccle and De Bilt have been given in great detail in Van Malderen et al. (2016), we include a reference to these findings here. We added the following text: "*The statistically insignificant offset between the Uccle and De Bilt trend estimates is*

*dependent on the used correction methods at both sites, but also differences in the vertical ozone distribution (up to 5% in the stratosphere), of both geophysical and instrumental origin, have an impact on the trend values (see e.g. Figs. 10a and 12 in Van Malderen et al. (2016), in which a more detailed explanation of the differences in vertical ozone distribution and trends between Uccle and De Bilt is given)."*

25. Lines 304-305 What level of "the stratosphere"?
    As the text has been changed significantly, following your next comment, this sentence has been replaced.

26. Lines 302-325 Unfortunately I find this discussion very "hand-wavy". There is a general discussion of results from the literature but I find it hard to discern what the specific conclusions are.
    Independently from our study, Ball et al. (2020), based on CCMs, also looked at the imprints of decrease ozone in the temperature variability, based on the same arguments. We included their findings in this discussion and confronted them with ours. We also want to draw your attention to the fact that the already cited Philipona et al. (2018) paper used the Uccle ozonesonde data, averaged together with the Payerne and Hohenpeissenberg soundings, to also study the link between stratospheric temperature and ozone trends. So, to our opinion, the literature discussion is not so general and really tied to our dataset as well. The same argument is true for the tropopause height variability: we first give the tropopause height variability at Uccle and De Bilt, and compare these trend estimates from a recent, global literature study. Then, we gave a general explanation of this global tropopause height variability, in which the Uccle and De Bilt cases fit. However, we rewrote this discussion in lines 302-325 substantially.

27. Lines 345-350 Frankfurt and Munich seem a long way from Uccle if considering boundary layer ozone – would you expect the trend to be correlated over this distance?
    Not for boundary layer ozone, but the focus is here merely on free-tropospheric ozone, for which a typical horizontal ozone correlation length is about 500 km (Liu et al., 2013). We added this reference to the text. To be more specific about boundary layer ozone, we added to the text "Near the surface, the De Bilt trend is in better agreement with the Frankfurt trend, but the local surface ozone production and destruction and the boundary layer dynamics can vary substantially between the three sites considered here, *so that the boundary ozone distribution and trends at the three sites are likely to be uncorrelated. However, comparing the lower-tropospheric IAGOS measurements at Frankfurt with nearby (within 50-80 km) and more distant (within 500 km) surface stations, Petetin et al. (2018) showed that the IAGOS observations in the first few hundred meters above the surface at Frankfurt airport have a representativeness typical of suburban background stations (like e.g. Uccle and De Bilt are), and as one moves higher in altitude, the IAGOS observations shift towards a regional representativeness."*

28. Line 342 (Figure 5) This is a very important figure but the way you've drawn it is unintentionally misleading. The reader's eye sees that the black curve and the red curve agree very well, and agree better than the blue, and it's easy to miss the fact the black curve is for a completely different time period. I would suggest only showing the three 1995-2018 trends on the one plot. If you want to include the 1969-1994 Uccle trend it could be a different panel.
    There is a clear legend in the figure, we think. Furthermore, we tried to use a consistent colour coding for different figures in the manuscript (Figs. 2, 3, S6): black for the entire Uccle period, green for the most recent Uccle sub-period (from 2000 for stratospheric ozone, from 1995 for tropospheric ozone), and blue for the De Bilt data.

We added this information in the figure caption and explicitly referred to the colour of the vertical ozone trends in the text. We think it makes sense to include those vertical trends in one panel, to make it visually easier to directly compare their magnitudes. We do not want to include the 1969-1994 Uccle trend here (the 1969-1996 tropospheric ozone trends are already present in Fig. S3).

29. Line 342 (Figure 5) Wouldn't a lot of the height range shown in this plot be above the troposphere at Uccle, particularly in winter?
The tropopause heights at Uccle and De Bilt vary, in the mean, between 10.5 km in winter months and 11.5 km in summer months (with standard deviations between 1 and 1.5 km). So, some of the height range shown in this plot is showing variability of stratospheric ozone as well. We included this information in the text when discussing the upper-tropospheric trends: "*Also, at those altitudes, the trends do not represent the tropospheric ozone time variability only, as the mean tropopause height range between 10.5 km (winter time) and 11.5 km (summer time), with standard deviations between 1 and 1.5 km, both at Uccle and De Bilt. As a consequence, lower-stratospheric ozone concentrations will contribute to the estimated trends in the upper altitude levels in Fig. 3.*"

30. Lines 399-403 This result seems extraordinary. Could the number of events really have increased by a factor of eight in only fifty years? It seems very implausible, and if you want to include it you need to show much more convincing evidence. (Otherwise the reader will assume the most likely explanation is that there's a mistake in your algorithm).
The result on the tropopause fold frequency increase is based on an algorithm, developed for ozonesonde profiles by Van Haver et al., 1996, that identifies tropopause folds based on 6 criteria (2 for ozone, and then 1 for relative humidity, stability, vertical gradient of wind speed and wind speed), see also Fig. S8 in the supplementary material for an visual illustration of those criteria in one profile. Only if those 6 criteria are fulfilled, a tropopause fold is identified. Van Haver et al. used this algorithm in automatic mode. The results shown in the manuscript are done in automatic mode as well. Here, two important remarks should be made:
1) As noted in the text, the higher vertical resolution of the sounding data in the more recent digital era (since 1990) might have an impact on the larger detected number of tropopause folds, although the amount of events has continuously increased since then. For instance, the increase in the digital era is 0.12±0.05% per year. For the first 25 years of the time series (until 1995), Van Haver et al. (1996) mentioned an increase of 0.07±0.06%, while we obtain an increase of 0.10±0.06% since 1995. So, the increase of the frequency seems to be independent of the vertical resolution of the sounding data, but the rate of increase might be affected by it.
2) We also used the algorithm in manual mode. This means that the vertical profiles of all relevant variables are plotted if at least 5 tropopause fold identification criteria are met. In this mode, one can ascertain visually if a tropopause fold is really present in the profile or not, and why criteria are met or not. In this manual/visual inspection mode, which is of course more subjective, more tropopause folds have been identified (344 instead of 290), and in particular in the beginning of the time series. The main reason is that the automatic method do not identify a tropopause fold when there is no relative humidity data, or when the humidity sensor was clearly iced at the tropopause fold location (following the icing recognition algorithm developed by Leiterer et al., 2005, doi:10.1175/JTECH-1684.1). Unfortunately, missing or iced humidity sensor data occur mostly for the older radiosonde humidity sensors (VIZ, Vaisala RS80). More recent radiosonde types in use in Uccle (since 2007: RS92 and RS41) do not suffer from iced sensors because they are equipped with 2 sensors, that are

heated alternately. As a result, the time series of manually identified tropopause folds shows a lower increase rate of the frequency, of about 0.09±0.02% per year for the entire 1969-2018 time period. After 1990, the increase rate amounts to 0.05±0.04% per year.

So, to conclude, to be as objective as possible in the tropopause fold identification, we prefer to use the automatic detection mode of the algorithm, but we point to a possible overestimation of the trend of the frequencies due to the reasons summed up here above. However, given the sensitivity analysis for those two impacting factors, we believe that the increase in the trend frequency is very robust and an important result (see references in the manuscript). Moreover, the very high vertical resolution of ozonesonde profiles (and of the coupled radiosonde profiles of meteorological variables), makes this one of the most suitable datasets for identifying tropopause folds. Finally, the (global) climatology of tropopause folds is recently gaining importance, as (chemical) reanalyses like CAMS and climate models are used to analyse this (e.g. recent work by Akritidis et al., 2019 and 2021, resp. doi: 10.5194/acp-19-14387-2019 and doi: 10.1029/2020JD034115). Currently, we are also working on a similar analysis of ERA5, but results are too preliminary to already share at this moment. Based on these arguments, we would argue to keep this paragraph on board of the paper, but we are open to omit it if you insist on it.

The manuscript has been changed as follows: "*On one hand, the large increase over the entire time period might be explained to some technical aspects. First, the higher vertical resolution of the sounding data in the more recent digital era (since 1990) might have an impact on the larger detected number of tropopause folds (thinner layers might be detected), although the amount of events has continuously increased since then, at a slightly smaller rate of 0.12 ± 0.05 % per year. Secondly, a visual inspection of all profiles fulfilling at least five of the tropopause fold detection criteria, led to a higher number of (manually) identified events (around 50), and (relatively) especially in the beginning of the time series. This is explained by the fact that the low humidity criterion was often not met in the automatic detection, because there were no humidity data or the humidity sensor was iced (following the icing recognition algorithm of Leiterer et al., 2005). More recent types of radiosonde humidity sensors (in use since 2007 at Uccle) prevent ice contamination by heating them during flight. However, this manual (and hence more subjective) mode of the algorithm still gives a 0.09 ± 0.02 % increase per year of the tropopause fold events since 1969. Therefore, we believe that the significant increase, although possibly overestimated by the automatic procedure, is nevertheless a robust feature of the analysis here. On the other hand, a higher rate of tropopause folding events is expected due to climate change (Tarasick et al., 2019, and references therein): climate change is projected to increase planetary wave activity and so cause an accelerated Brewer-Dobson circulation. This acceleration, along with stratospheric ozone recovery, will lead to increased transport of ozone from the stratosphere into the troposphere and hence more tropopause folding events. Akritidis et al. (2019) add to this that the degree of increase in the downward transport of stratospheric ozone is partially driven by the long-term changes in tropopause fold activity.*"

31. Lines 412-465 Unfortunately, I have to say my honest recommendation would be to delete section 4.4 altogether. For one thing, it is not clear that the ozonesonde record adds any benefit compared to the continuous surface monitoring. (Noting that the ozonesonde data before 1985 is of limited value). There is a small offset which you don't seem to explain. Secondly, although you show an increasing trend, after a fairly lengthy discussion you are not able to reach any conclusions about its causes (in terms of changes in the precursors or meteorological conditions or some other factor).

We moved section 4.4 to an appendix. We really think that this section should not be deleted altogether. Concerning your first point: we do not claim that the ozonesonde

record adds any benefit compared to the continuous surface monitoring. As a matter of fact, it is the other way around: the agreement in trend with the surface monitoring device gives some credibility to the ozone sounding measurements near the ground, despite the impact of pre-launch procedures on the surface ozone measurements, which might be a reason for the small offset as well (explained in the text now). These two points have been raised respectively in the text as "*In this appendix, we elaborate more on how representative and complementary the surface ozone trend derived from the ozonesonde data at Uccle is, compared to the one from a surface station at the same site. The ground network of (air quality) stations provides surface ozone measurements at higher temporal and horizontal resolution, and with higher accuracy than ozonesonde measurements, but these latter provide vertical ozone profiles in the lower troposphere as well, and sometimes even over a longer time span.*" & "*This offset might be explained by the difference in air masses for which the ozone concentrations are measured (surface vs. surface to 1 km above the ground), and by some Uccle pre-launch procedure of testing the ozonesonde-interface-radiosonde configuration by exposing the ozonesonde shortly (< 30 s) to (stratospheric) ozone concentrations between 15 to 30 minute prior to launch. Because of the slow time constant of 20-25 minutes in the chemical reactions in the cell, this pre-launch ozone exposure might still contribute to the measured cell current after launch, resulting in a positive bias in the boundary layer ozone measurements with the ozonesondes.*"

Concerning your second point, we agree that the discussion about the trends was rather lengthy and perhaps a bit too general. So we shortened and reorganized the discussion by concentrating on the ozone (mean) trends (only since 1986 for ozonesonde data as well) first, describe the general possible causes for them, and then concentrate on the link with the ozone precursor emission changes, because these measurements are available at the Uccle site (or nearby for CO). Finally, we describe the Uccle surface ozone trends for the lowest and highest percentile data (Figure S10 in Supplementary Material), as these give additional information for explaining the increase in (mean) surface ozone amounts. Finally, we also want to draw your attention to the fact that a clear identification of which factors contribute most to the observed surface ozone trends is not straightforward at any site and has consequently hardly been for any site in the literature so far. We think that the summarizing conclusion contains it all: "*To conclude, explaining the increasing mean surface ozone amounts in combination with the decreasing ozone precursor emissions at Uccle is less straightforward than the (opposing) trends in high and low level ozone concentrations due to the compression of the surface ozone distribution. The interpretation of the increasing mean surface ozone concentrations is hampered by the interplay of many factors such as meteorology and transport, the non-linear dependence of the ozone concentrations on the emissions of VOC and NOx, the dual role of NOx as ozone source or sink depending on the season, and the amount of NOx emissions.*"

32. Line 464 (Figure 7) Overall the figure is cluttered and difficult to read. Just from looking at the plot, while there appears to be a good agreement between the ozonesonde and the surface monitor, it is really just the existence of a consistent seasonal cycle. There seems to be a jump around the year 1998?
We decided to replace this Figure 7 by two figures. One figure shows only the surface ozone monthly means from ozonesondes and the surface monitor, so that the offset and the consistent seasonal cycle is highlighted. In the other figure (inspired by a similar figure that was originally in the supplementary material), we show the monthly anomalies of the surface ozone by the two devices (we do not identify a jump around 1998 in this figure and it should be clear that both time series agree also rather well after removal of the seasonal cycle), with in the lower panels

the monthly anomalies from the co-located NO, NO$_2$ and CO concentrations. All monthly anomaly time series do not start earlier then 1986 and the figure has been enlarged as well, so we hope the figure is less cluttered and easier to read now.

33. Line 484 Who is "us" in this context?
   We removed "us" here.

34. Line 503 You should explain why applying the averaging kernels makes a much larger improvement in the lower stratosphere compared to other altitudes.
   Yes, you are absolutely right. This is because the lower stratosphere is the region with the highest ozone variability, so that, smoothing the high vertical resolution ozonesonde data to correctly inter-compare both products at the same vertical resolution, will have the largest effect here by removing the details of the differences.
   In the text, we added: "*The lower stratosphere is the region with the highest ozone variability, so smoothing the high resolution ozonesonde profiles to the GOME-2 vertical resolution will have the largest effect here by removing details of the differences.*"

35. Lines 505-509 This would be more convincing if you didn't just show the "before" and "after" plots – the reader is left wondering whether the degradation correction has been tuned to match the ozonesondes – perhaps you could show the degradation correction over time?
   Thank you for your comment. The degradation correction has certainly not been tuned to match the ozonesondes. We gave in the manuscript some more information about the degradation and its correction: "*For example, the measured values of the GOME-2A Irradiance in the UV (below 300nm) has reduced by roughly 80% in 2016 (since its launch in 2007). Since the vertical ozone profile retrieval algorithm depends on an absolute calibrated reflectance (sun normalised radiance) there is a need to correct for this temporal change of the (joint) radiance and irradiance. This method depends on the assumption that, taken as an average across the globe, the atmospheric constituents (mainly ozone) will be close to the multiyear climatological value from McPeters and Labow (2012). The climatological ozone profile is then scaled with the Assimilated Total Ozone columns to get the overall ozone absorption correct (Tuinder et al., 2019).*"
   Figure 6.3 in the document shows the effects of the different methods that can be applied to re-distribute an excess or shortcoming of the expected ozone column.
   The relative differences between the GOME-2A retrievals and NH midlatitude ozonesonde ozone measurements after applying the degradation error correction (Fig. 5) or not (Fig. 6) are presented in the plots here below. It can be seen that the agreement between both datasets is better, especially at but not restricted to the end of the time series when the degradation error correction is applied. However, we believe that this is out of scope for the present manuscript that focuses on the Uccle dataset, as the validation of this degradation error correction requires a more global approach (as shown in the plots here below).

[Figure]

**Figure 5: Time series of relative ozone differences between GOME-2A and NH midlatitude ozonesonde at 6 different altitude levels for January 2013 - December 2018 for the GOME-2A time series that have been corrected for the degradation error.**

[Figure]

**Figure 6: Time series of relative ozone differences between GOME-2A and NH midlatitude ozonesonde at 6 different altitude levels for January 2013 - December 2018 for the GOME-2A time series that have been not been corrected for the degradation error (operational processing).**

36. Line 531 In section 5.2 you don't use the averaging kernels but in section 5.1 you said how important it was to apply them – why is this?

It really depends on the satellite ozone retrieval technique with which the ozonesondes are compared. In the case of MLS, "the relatively high vertical resolution due to the limb sounding technique, allows for many scientifically useful studies to be undertaken without reference to the averaging kernels" (see pag 7, Livesey et al., 2020). Moreover, from Fig. 3.18.8 (top) on page 128 in the same document, the dashed line (the FWHM of quasi-triangular functions) shows that the vertical resolution of MLS is rather stable around 2.5-3 km between 10-200 hPa (vertical range of interest for comparison with Uccle ozonesondes). So, we simulated those averaging kernels by determining around each MLS pressure level a pressure range that coincides with half of this vertical resolution. For this pressure range, we then interpolated the ozone concentrations in the ozonesonde profile to the MLS pressure level, and this value was then compared with the MLS measurement at that pressure level. This interpolation is to be preferred above layer averaging (see again

Livesey et al., 2020), but the differences between both approaches for Uccle are at most 8% at 200 hPa, and less than 3% between 100-10 hPa. Another drawback of using the averaging kernels for MLS is that MLS provides only one fixed averaging kernel for our latitude for all data, which is e.g. not the case for GOME-2 and TES.

We modified the text into: "*Thanks to the relatively dense and regular MLS vertical resolution of around 2.5 km in the 10-200 hPa pressure range, it is feasible to interpolate the Uccle ozonesonde data to the MLS pressure levels on a fine pressure grid of 2.5 km. Applying the time invariant MLS averaging kernel on the latitude of Uccle on the ozonesonde data did not have a large effect on the smoothing of the vertical ozonesonde profile, as compared to applying the identify matrix to the ozonesonde vertical profile (< 1%). This contrasts strongly with the GOME-2 and TES retrievals (see Sect. 5.3), where the spatio-temporal varying averaging kernels affect the vertical ozone profiles substantially, and as such should be used on the sonde data for pairwise comparison.*"

37. L 537-538 The figure shows a growing bias at altitudes above 10 hPa – it seems to be a systematic effect and not just that the ozonesondes are "known to be less accurate"

Yes, it a systemic effect, in the ozonesonde measurements. You are completely right. Above 10 hPa, the sensing solutions either boil/evaporate and freeze, changing completely the stoichiometry of the chemical reactions, but mostly leading to an underestimation of the true ozone concentrations. However, at those pressure levels, the pump efficiency decrease of the ozonesonde is really large, with large uncertainties, which makes the ozone measurement not very accurate there as well. Both effects increase with increasing altitudes. Above 10 hPa, it is therefore no longer advised to use ozonesonde data, as simulation chamber experiments with a photometer as reference have shown (Smit et al., private communication). The same effect is seen in the composite ECC-MLS comparisons in Fig. 3 of Stauffer et al. (2020), for both manufacturers of ECC ozonesondes.

The text has been changed into "*At pressures smaller than 10 hPa, ozonesonde measurements are systematically underestimating ozone due to the evaporation or freezing of the sensing solutions (see also the composite ECC-MLS Fig. 3 in Stauffer et al., 2020), and they have a larger uncertainty due to increased pump efficiency uncertainty at low pressures.*"

38. Lines 545 (Figure 9) Overall I like this figure but the error bars seem strange. If the agreement from one year to the next is so good, it doesn't seem possible that the error bars could really be so large. The description in the caption is too brief and doesn't show what the error bars are really representing.

The caption has been changed into "*Relative ozone profile differences between MLS and Uccle ozonesondes. The different colours correspond to the different yearly averages, illustrating the large consistency among those. The black line represents the overall mean relative differences, with the error bars the one standard deviations due to the individual differences. Therefore, it could be noted that individual differences are relatively large at some pressure levels, but they are cancelled out in the yearly mean.*"

39. Line 551 Here, you should also let the reader know that TES was decommissioned in early 2018.

Thank you! We added this information.

40. Line 552 You have already described the orbit of Aura in the MLS section.

Thank you, we removed the Aura orbit description here.

41. Line 559 It seems curious that you apply a limit of 300 km in the troposphere but 100km in the stratosphere – is this a reasonable thing to do?

It is common practice to use 300 km between the sonde launch location and the overpass of the satellite to compare ozone profiles in the troposphere for TES (see Nassar et al., 2008; Verstraeten et al., 2013, among others) to ensure a reasonable amount of pairs of ozone profiles. When we use the 100 km criteria we only have 33 pairs, otherwise 191, which is a substantial difference. For tropospheric profiles, the quality flag is determined by clouds (especially for Uccle which has many cloudy and rainy days) and retrieval errors, among others, reducing the amount of data passing the quality flag.

42. Line 563 You should explain to the reader what is meant by the term "observation operator". Is it the same as an averaging kernel?

Yes, indeed, the observation operator is the same as the averaging kernel. In order not to mix terminology we have changed it in the revised manuscript to averaging kernel.

43. Line 580 In this case, seeing there are only "1-2" degrees of freedom in the troposphere, wouldn't it better to make a plot of the regression described in lines 581-590, rather than figure 10 as it currently stands? Then the reader could see the temporal stability.

We agree. We have added the correlation plots for the lower and upper troposphere in the revised manuscript, as lower panels in what becomes Figure 7 now.